# Mortality predictors, hepatic involvement patterns, and the steatotic liver paradox in 1,484 hospitalized Dengue patients

Aryalakshmi Sreemohan[1], Arif Hussain Theruvath[1], Ambily Baby[1], Cyriac Abby Philips [1,2*], Tharun Tom Oommen[2,3], Santhichandra Pai[4], Salini Baby John[4], Jaicob Varghese[5], Rizwan Ahamed[3], Ajit Tharakan[3], Philip Augustine[3]

**1** Clinical Research Division, The Liver Institute, Center of Excellence in Gastrointestinal Sciences, Rajagiri Hospital, Aluva, Kerala, India, **2** Department of Clinical and Translational Hepatology, The Liver Institute, Center of Excellence in Gastrointestinal Sciences, Rajagiri Hospital, Aluva, Kerala, India, **3** Department of Gastroenterology and Advanced GI Endoscopy, Center of Excellence in Gastrointestinal Sciences, Rajagiri Hospital, Aluva, Kerala, India, **4** Department of General (Internal) Medicine, Rajagiri Hospital, Aluva, Kerala, India, **5** Department of Intensive Medicine and Critical Care, Rajagiri Hospital, Aluva, Kerala, India

* abbyphilips@theliverinst.in

## Abstract

### Background and objectives

Dengue fever represents a major global health burden with hepatic involvement occurring in up to 90% of hospitalized patients. This study aimed to determine in-hospital mortality predictors, characterize clinical outcomes across graded hepatic injury phenotypes, and validate prognostic scoring systems in hospitalized dengue patients. A primary focus was assessing the impact of pre-existing liver conditions on disease trajectory and survival.

### Methods

A retrospective cohort analysis was conducted on 1,484 patients with laboratory-confirmed dengue infection admitted to a tertiary-care center between 2021 and 2024. Patients were stratified by hepatic status into those with pre-existing chronic liver disease, non-chronic steatotic liver involvement, and no liver involvement. Multivariable logistic regression identified independent mortality predictors, while unsupervised clustering distinguished clinical phenotypes. The performance of physiological and liver-specific prognostic scores was evaluated against clinical outcomes.

### Results

The overall in-hospital mortality rate was 5.1%, with 13.1% requiring intensive care admission. Independent predictors of mortality included severe dengue classification, intensive care unit admission, elevated neutrophil-to-lymphocyte ratio, and hypoalbuminemia, with albumin emerging as the strongest single biomarker for risk prediction.

**Data availability statement:** All relevant data are within the manuscript.

**Funding:** The author(s) received no specific funding for this work.

**Competing interests:** The authors have declared that no competing interests exist.

Paradoxically, patients with steatotic liver disease demonstrated improved survival compared to those without pre-existing liver disease, supporting an "obesity paradox" in this tropical infection context, whereas decompensated cirrhosis was associated with markedly adverse outcomes. The Albumin-Bilirubin grade successfully stratified hepatic risk, and the Simplified Acute Physiology Score-3 significantly outperformed the Sequential Organ Failure Assessment for predicting mortality in critically ill patients. Four distinct clinical phenotypes with differential mortality ranging from 3.0% to 100% were identified.

## Conclusions

Hypoalbuminemia serves as a critical, accessible prognostic marker in dengue fever. Pre-existing liver pathology demonstrates divergent impacts on outcomes, with steatotic liver disease potentially conferring survival advantage contrary to traditional metabolic risk assumptions. These findings support the utility of liver-specific scoring systems for acute risk stratification in dengue-endemic regions.

## Introduction

Dengue has transcended its historical characterization as a self-limiting tropical fever to become one of the most rapidly expanding arboviral infections worldwide, with the World Health Organization (WHO) reporting unprecedented case surges that have strained healthcare systems across endemic regions. Recent WHO updates underscore the scale and volatility of dengue transmission, with explosive year-on-year increases in reported cases in many endemic regions and recurring surges that strain hospital and ICU capacity [1]. The global burden continues to escalate, with an estimated 390 million infections annually, of which approximately 96 million manifests clinically. This expansion is driven by urbanization, climate change, and increased human mobility, creating conditions favourable for sustained transmission. The clinical spectrum of dengue ranges from mild febrile illness to severe dengue characterized by plasma leakage, haemorrhage, and organ dysfunction, as codified in the WHO 2009 classification framework that continues to guide bedside triage [2].

Among organ complications, hepatic involvement is particularly common and clinically heterogeneous. Aminotransferase elevations occur in 30–90% of hospitalized patients, with aspartate aminotransferase (AST) characteristically exceeding alanine aminotransferase (ALT). While most patients exhibit only mild transaminase derangements, a minority progress to acute liver injury, fulminant hepatic failure, or liver dysfunction as part of multi-organ failure – clinical states that carry disproportionate mortality [3]. Mechanistic pathways underlying hepatic injury include direct viral cytopathic effects, immune-mediated hepatocyte damage, systemic inflammatory responses, and hypoxic-ischemic injury in the context of shock; in practice, these processes often overlap and are difficult to disentangle at the bedside [4].

These clinical outcomes assume greater relevance in patients with pre-existing liver disease or chronic liver disease (CLD). Cirrhosis may amplify dengue severity through reduced hepatic reserve, baseline coagulopathy and thrombocytopenia, cirrhosis-associated immune dysfunction, and heightened susceptibility to infection and organ failure. However, the evidence base remains fragmented – often limited to smaller cohorts, heterogeneous definitions of liver disease severity, and minimal stratification by liver disease severity status [5]. A systematic review and metanalysis demonstrated that liver cirrhosis was associated with increased hospitalization, intensive care unit (ICU) admission, and mortality in dengue patients, yet granular clinical characterization was lacking [6]. Similarly, while hepatic manifestations have been well-described in dengue, the specific impact of CLD on clinical trajectories and outcomes remains inadequately characterized.

Concurrently, metabolic dysfunction-associated steatotic liver disease (MASLD) has emerged as the most prevalent chronic liver condition globally, affecting approximately 30% of adults [7]. Traditional teaching identifies metabolic syndrome as a risk factor for severe dengue, positing that chronic pro-inflammatory states may exacerbate cytokine storm [8,9]. However, emerging evidence from critical care medicine introduces the concept of the "obesity paradox," wherein increased adiposity may paradoxically confer survival advantage in acute critical illness through enhanced metabolic reserves, higher leptin and adiponectin levels, or modulated inflammatory responses [10–13]. Whether this phenomenon extends to tropical viral infections – where thrombocytopenia is universal and organ failure kinetics differ from bacterial sepsis – remains unexplored. The limited available literature on steatotic liver disease and dengue outcomes yields contradictory signals, potentially confounded by variable admission thresholds and metabolic comorbidity burden.

In the face of rising caseloads and increasingly complex patient phenotypes, accurate prognostication is paramount for triage and resource allocation. Traditional critical care scoring systems like the Sequential Organ Failure Assessment (SOFA) and Simplified Acute Physiology Score (SAPS) were developed primarily for bacterial sepsis in Western populations [14]. Their applicability to tropical viral infections, where thrombocytopenia is universal and organ failure kinetics differ from bacterial sepsis, remains under-validated. Furthermore, liver-specific scoring systems such as the Albumin-Bilirubin (ALBI) score, originally designed to assess hepatic reserve in hepatocellular carcinoma (HCC) without the subjective confounders of the Child-Pugh score (ascites, encephalopathy), offer a potential objective tool for acute risk stratification. The validation of ALBI, alongside comparisons of SAPS-3 versus SOFA in a large dengue cohort, represents a critical unmet need in tropical medicine [15].

Against this backdrop, we conducted a single-center retrospective cohort study analysing a large database of hospitalized dengue patients, explicitly stratifying outcomes by hepatic status: pre-existing CLD, non-CLD liver involvement (predominantly steatotic liver disease), and no liver involvement. We further aimed to characterize outcomes across graded hepatic injury phenotypes through validated multiple prognostic scoring systems. The primary objective was to determine in-hospital mortality and its predictors. Secondary objectives included ICU admission, length of stay, readmission patterns, longer-term survival, and identification of clinically usable predictors of adverse outcomes including novel phenotypic characterization through unsupervised clustering approaches. We hypothesized that pre-existing liver disease would be associated with worse outcomes, that current prognostic scores would demonstrate variable performance in this tropical infection context, and that distinct patient phenotypes with differential mortality could be identified through machine learning approaches.

## Patients and methods

### Study design and population

This retrospective cohort study analysed all patients with Dengue (NS1 antigen-positive or serologically confirmed) infection admitted to our institution between April 2021 and August 2024. Data were extracted from electronic medical records (initiated on 15th December 2025) including demographics, clinical characteristics, laboratory parameters, in-hospital events, and follow-up outcomes. Baseline characteristics collected included age, sex, body mass index (BMI), and comorbidities (diabetes mellitus, hypertension, dyslipidaemia, hypothyroidism, cardiac disease, asthma/COPD, and

chronic kidney disease). Clinical variables included Dengue classification (classical, severe, severe with shock), presence of haemorrhagic manifestations, shock at presentation, ascites, pleural effusion, hepatic encephalopathy, and requirement for mechanical ventilation or dialysis. Laboratory parameters included complete blood count, liver function tests, renal function tests, coagulation profile, and inflammatory markers (CRP, ferritin). This retrospective study was approved by the Institutional Ethics Committee of Rajagiri Hospital (Study Reference No. RAJH/2024/090, dated 29.09.2024). Given the retrospective nature of the study using de-identified data from electronic medical records, the requirement for individual informed consent was waived. All analyses were performed in accordance with the Declaration of Helsinki and institutional policies for retrospective research. Patient confidentiality was maintained throughout the study, with all identifying information removed prior to analysis.

## Definitions and classifications

Patients were classified according to WHO 2009 criteria into: classical dengue (dengue without warning signs), severe dengue (dengue with clinically relevant plasma leakage, severe haemorrhage, or severe organ dysfunction), and dengue with shock (dengue shock syndrome with hypotension).

Hepatic involvement was categorized as: abnormal liver tests (aspartate aminotransferase, AST or alanine aminotransferase, ALT 1–3 × upper limit of normal, ULN), acute hepatitis (AST or ALT 3–10 × ULN), acute hepatitis with jaundice (any abnormal elevation of AST or ALT with bilirubin ≥3 mg/dL), acute liver injury (ALI = acute hepatitis with jaundice and raised international normalized ratio, INR ≥ 1.5), and liver failure as part of multi-organ failure (MOF, hepatic failure with concurrent organ failures). Non-CLD steatotic liver disease was diagnosed based on abdominal ultrasonography findings (increased liver echogenicity, hepatorenal contrast, and/or posterior beam attenuation) documented during or prior to the index hospitalization, as reported in formal radiology reports within the electronic medical records.

Secondary hemophagocytic lymphohistiocytosis (HLH) was defined using modified HLH-2004 criteria with emphasis on hyperferritinemia (>500 ng/mL) in the context of cytopenias and clinical criteria based deterioration.

Acute kidney injury (AKI) was operationally defined as serum creatinine >1.5 mg/dL at admission, as baseline creatinine values were unavailable for most patients, precluding application of KDIGO criteria. This simplified definition should be interpreted as renal impairment at presentation. Chronic kidney disease (CKD) was defined as pre-existing renal disease documented in the medical history at admission or on imaging.

ALBI score, representing liver disease severity was calculated as: (log10 bilirubin × 0.66) + (albumin × −0.085). ALBI grades were defined as: Grade 1 (≤−2.60), Grade 2 (>-2.60 to ≤-1.39), and Grade 3 (>-1.39). Sequential Organ Failure Assessment (SOFA, https://clincalc.com/icumortality/sofa.aspx) and Simplified Acute Physiology Score-3 (SAPS-3, https://www.evidencio.com/models/show/1114) were calculated for ICU patients using standard validated criteria based calculators.

The primary outcome was in-hospital mortality. Secondary outcomes included ICU admission, hospital length of stay, readmission after discharge, and long-term survival at follow-up. Patients were classified into three liver involvement groups: (1) Pre-existing CLD, (2) Non-CLD liver involvement (primarily steatotic liver disease), and (3) No liver involvement and similar sub-groups were created with respect to the variable and outcome analysed.

## Statistical analysis

Continuous variables were expressed as mean ± standard deviation (SD) or median with interquartile range (IQR) based on distribution normality. Categorical variables were presented as frequencies and percentages. Comparisons between groups were performed using Kruskal-Wallis test for continuous variables and chi-square test or Fisher's exact test for categorical variables.

Univariate logistic regression was performed to identify potential predictors, with variables achieving p < 0.1 entered into multivariable models. Multivariable logistic regression models were constructed using stepwise selection with entry

criterion p < 0.05 and removal criterion p > 0.10. Model performance was assessed using the concordance statistic (C-statistic/AUC). Adjusted odds ratios (aOR) with 95% confidence intervals were reported. Model calibration was assessed using the Hosmer-Lemeshow test. Multicollinearity was assessed using variance inflation factors (VIF).

Receiver operating characteristic (ROC) curves were constructed for continuous biomarkers. Optimal cutoff values were determined using Youden's J statistic (sensitivity + specificity – 1). Area under the ROC curve (AUC) with 95% confidence intervals was calculated using the DeLong method. Where appropriate, results were expressed as odds ratios (OR) with 95% confidence intervals (CI). Model discrimination was assessed using the C-statistic (area under the receiver operating characteristic curve).

Survival analysis was performed using Kaplan-Meier method with log-rank test for comparisons. Cox proportional hazards regression was used to identify factors associated with mortality, with results expressed as hazard ratios (HR) with 95% CI.

Year-over-year trends were assessed using Spearman correlation coefficients for ordinal/continuous outcomes and chi-square test for trend for categorical outcomes. Seasonal patterns were analysed by categorizing admissions into pre-monsoon (March-May), monsoon (June-September), post-monsoon (October-November), and winter (December-February) periods.

K-means clustering was performed to identify distinct patient phenotypes. The optimal number of clusters was determined using the elbow method (within-cluster sum of squares) and silhouette scores.

Missing data were handled using complete case analysis for the primary regression models. Complete case analysis was performed for primary regression models, with sample sizes specified for each analysis. Sensitivity analyses confirmed that findings were robust to missing data patterns. The proportion of missing data for each variable was reported. Sensitivity analyses were performed to assess the impact of missing data on key findings. Variables with >20% missing values were excluded from multivariable models unless clinically essential.

Model performance was evaluated using ROC curve analysis with AUC and 5-fold cross-validation. Optimal biomarker cutoffs were determined using Youden's index. Three machine learning algorithms (Random Forest, Gradient Boosting, and Logistic Regression) were compared using cross-validated AUC scores. Variable importance was extracted from the Random Forest model.

Biological interaction was assessed using multiple approaches. Stratified analysis calculated stratum-specific ORs for the exposure-outcome relationship within levels of the potential effect modifier. The ratio of ORs was used to assess interaction on the multiplicative scale. Formal statistical testing used multivariable logistic regression with product interaction terms. A ratio of ORs > 1 suggests synergistic (supra-additive) interaction, while <1 suggests antagonistic (sub-additive) interaction.

All analyses were performed using Python 3.12 with the following packages: pandas (v2.0) for data handling, scipy (v1.11) for statistical tests, statsmodels (v0.14) for regression analyses, scikit-learn (v1.3) for machine learning algorithms and cross-validation, and matplotlib/seaborn for visualization. A two-sided p-value <0.05 was considered statistically significant. The proportion of missing data for each key analytical variable is reported in Supporting Information (S1 Text). Variables with >20% missingness were excluded from primary multivariable models. All variables included in the primary mortality and ICU models had < 5% missing data.

## Results

### (i) Study population and baseline characteristics

A total of 1,484 patients (Table 1) were included, with a median age of 44 years (IQR 28–58) and a male predominance (n = 831, 56%) (Fig 1). Case volumes increased significantly over the study period, rising from 63 patients (4.2%) in 2021– 763 (51.4%) in 2023 and 503 (33.9%) in the first half of 2024. The prevalence of comorbidities included diabetes mellitus

**Table 1. Baseline demographics and clinical characteristics.**

| Variable | Value |
|---|---|
| **Demographics** | |
| Age, years, median (IQR) | 44 (28-58) |
| Male sex, n (%) | 831 (56.0) |
| Age<18 years, n (%) | 159 (10.7) |
| Age 18–40 years, n (%) | 465 (31.3) |
| Age 40–60 years, n (%) | 521 (35.1) |
| Age ≥ 60 years, n (%) | 339 (22.8) |
| **Comorbidities** | |
| Diabetes mellitus, n (%) | 288 (19.4) |
| Hypertension, n (%) | 277 (18.7) |
| Dyslipidaemia, n (%) | 194 (13.1) |
| Cardiac disease, n (%) | 112 (7.5) |
| Chronic kidney disease, n (%) | 17 (1.1) |
| Pre-existing liver disease, n (%) | 34 (2.3) |
| COPD/Asthma, n (%) | 43 (2.9) |
| Any comorbidity, n (%) | 533 (35.9) |
| **Clinical Presentation** | |
| Classical dengue, n (%) | 977 (65.8) |
| Severe dengue, n (%) | 470 (31.7) |
| Shock dengue, n (%) | 37 (2.5) |
| Severe/Shock dengue, n (%) | 507 (34.2) |
| Shock at presentation, n (%) | 18 (1.2) |
| Haemorrhagic manifestations, n (%) | 202 (13.6) |
| Ascites, n (%) | 146 (9.8) |
| Pleural effusion, n (%) | 138 (9.3) |
| Secondary HLH, n (%) | 57 (3.8) |
| **Laboratory Parameters** | |
| Hemoglobin, g/dL, median (IQR) | 13.5 (12.3-14.8) |
| WBC, × 10³/µL, median (IQR) | 3.7 (2.7-5.3) |
| Platelets at admission, × 10³/µL, median (IQR) | 120 (63-159) |
| Platelet nadir, × 10³/µL, median (IQR) | 75 (36-120) |
| AST, U/L, median (IQR) | 74 (43-141) |
| ALT, U/L, median (IQR) | 58 (34-114) |
| Albumin, g/dL, median (IQR) | 4.0 (3.8-4.3) |
| Creatinine, mg/dL, median (IQR) | 0.9 (0.7-1.1) |
| Total bilirubin, mg/dL, median (IQR) | 0.6 (0.4-0.8) |
| INR, median (IQR) | 1.1 (1.0-1.2) |
| CRP, mg/L, median (IQR) | 9.1 (4.0-22.1) |
| Ferritin, ng/mL, median (IQR) | 1132 (387-2780) |
| NLR, median (IQR) | 2.0 (1.0-3.8) |
| **Hospital Course** | |
| ICU admission, n (%) | 194 (13.1) |
| Mechanical ventilation, n (%) | 17 (1.1) |
| Renal replacement therapy, n (%) | 8 (0.5) |
| Secondary bacterial infection, n (%) | 24 (1.6) |

*(Continued)*

**Table 1.** (Continued)

| Variable | Value |
|---|---|
| In-hospital mortality, n (%) | 75 (5.1) |
| Hospital stays, days, median (IQR) | 4 (3-6) |

*Continuous variables presented as median (IQR). Categorical variables presented as n (%).*

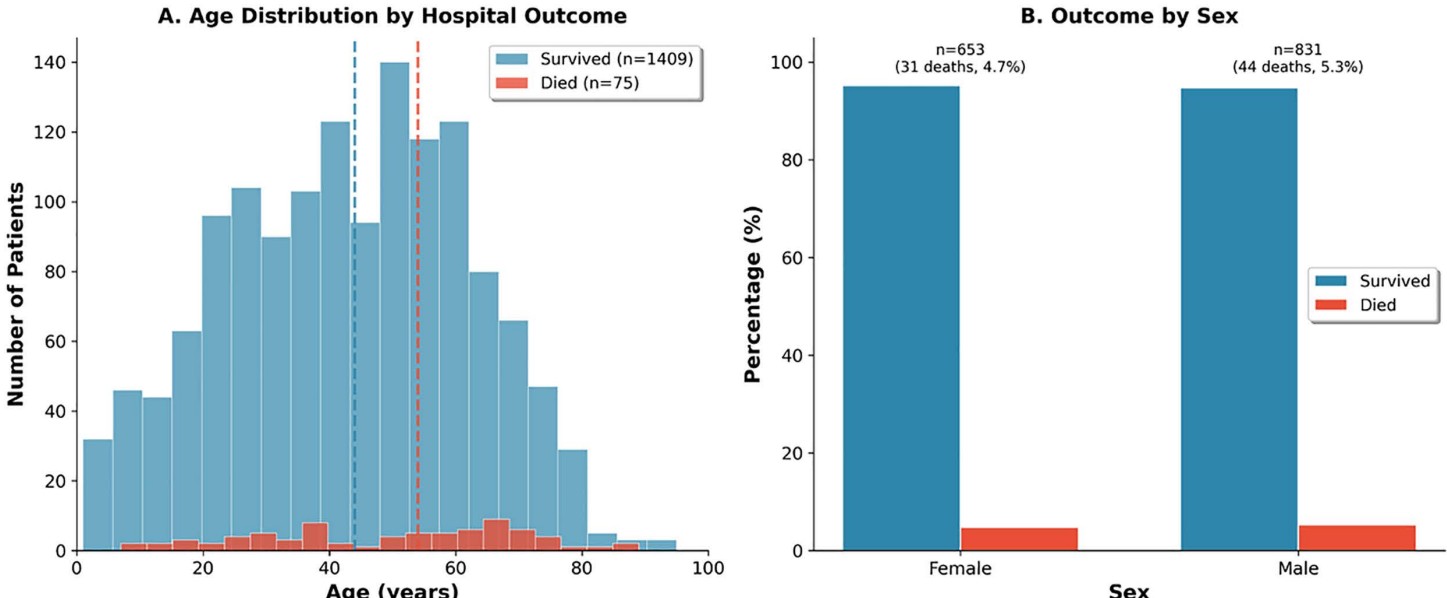

**Fig 1. Study population demographics. Panel A:** Age distribution histogram by hospital outcome. Died (n = 75): median 54 years. Survived (n = 1,409): median 44 years. **Panel B:** Outcome by sex. Males: 44/831 (5.3%) mortality. Females: 31/653 (4.7%) mortality.

(19.4%), hypertension (18.7%), and cardiac disease (7.5%). Clinically, 65.8% presented with classical dengue, 31.7% with severe dengue, and 2.5% with dengue shock (Fig 2).

### (ii) Primary and secondary outcomes

The median hospital length of stay was 4 days (IQR 3–6), with a range of 1–40 days. ICU admission was required in 194 patients (13.1%), mechanical ventilation in 17 (1.1%), and dialysis in 8 (0.5%). Among ICU patients (n = 194), median ICU stay was 3 days (IQR 2–4). In-hospital mortality occurred in 75 patients (5.1%) (Fig 3).

### (iii) Gender differences in clinical outcomes

Mortality rates were comparable between sexes (males 5.3% vs females 4.7%, p = 0.72). However, males demonstrated significantly higher rates of ICU admission (13.0% vs 8.6%, OR 1.59, 95% CI: 1.13–2.24, p = 0.009) and severe thrombocytopenia (12.0% vs 6.3%, OR 2.04, 95% CI: 1.40–2.98, p < 0.001). Haemorrhagic manifestations showed a trend toward higher rates in females (15.6% vs 12.0%, OR 0.74, 95% CI: 0.55–0.99, p = 0.054). Acute kidney injury, defined as creatinine >1.5 mg/dL at admission, occurred in 4.5% of males and 2.9% of females without significant difference (OR 1.55, 95% CI: 0.89–2.73, p = 0.158).

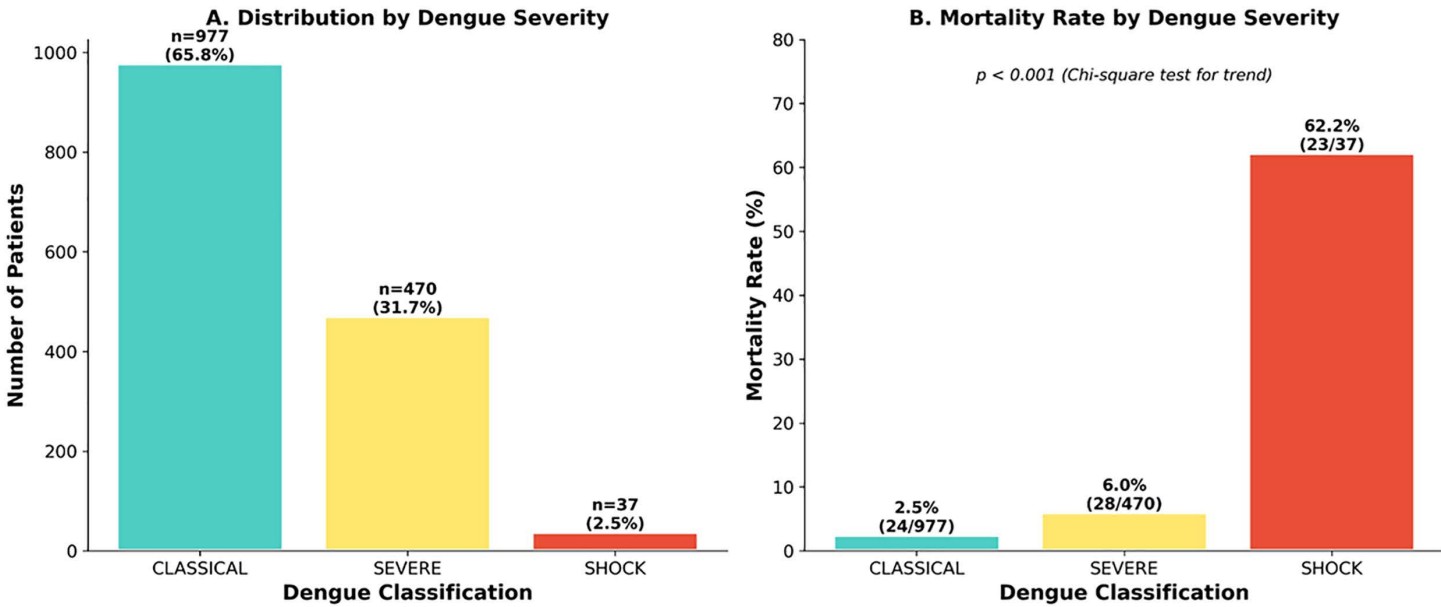

**Fig 2. Dengue severity classification and outcomes. Panel A:** WHO 2009 classification: Classical 977 (65.8%), Severe 470 (31.7%), Shock 37 (2.5%). **Panel B:** Mortality: Classical 24/977 (2.5%), Severe 28/470 (6.0%), Shock 23/37 (62.2%). Chi-square p < 0.001.

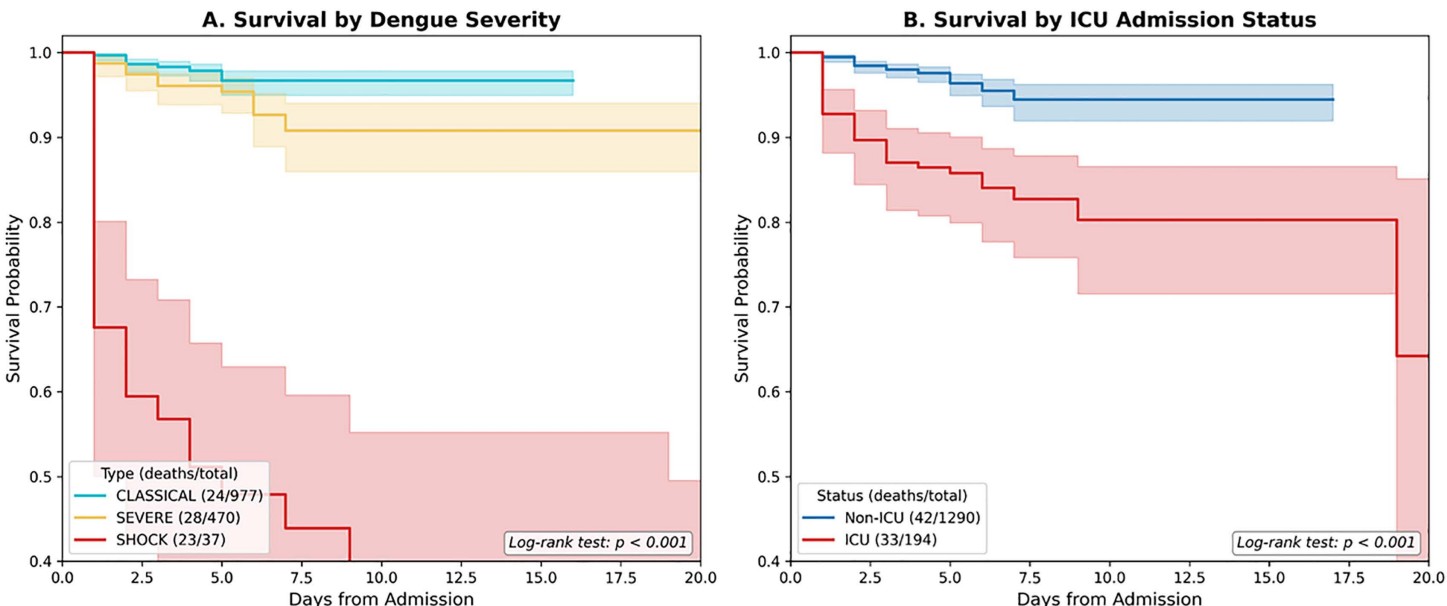

**Fig 3. Kaplan-Meier survival curves stratified by dengue severity and ICU admission status. Panel A:** Survival probability by dengue severity classification. Patients with dengue shock syndrome exhibited markedly worse survival compared to severe dengue without shock and classical dengue (CLASSICAL: 24/977 deaths, 2.5%; SEVERE: 28/470 deaths, 6.0%; SHOCK: 23/37 deaths, 62.2%; log-rank p < 0.001). **Panel B:** Survival probability by ICU admission status. ICU patients (admitted at presentation or transferred during hospitalization) had significantly higher mortality than non-ICU patients (ICU: 33/194 deaths, 17.0%; non-ICU: 42/1,290 deaths, 3.3%; log-rank p < 0.001). Shaded areas represent 95% confidence intervals. Total cohort: n = 1,484; 75 deaths (5.1% overall mortality).

## (iv) Metabolic syndrome and outcomes

Metabolic syndrome, defined as the presence of one or more of diabetes mellitus, hypertension, and dyslipidaemia, was present in 226 patients (15.2%). Patients with metabolic syndrome did not demonstrate significantly higher mortality (6.2% vs 4.8%, OR 1.30, 95% CI: 0.71–2.36, p = 0.493) or ICU admission rates (13.3% vs 10.7%, OR 1.28, 95% CI: 0.84–1.96, p = 0.297) compared to those without. However, metabolic syndrome was significantly associated with secondary bacterial infection (3.5% vs 1.3%, OR 2.85, 95% CI: 1.20–6.74, p = 0.021).

## (v) Predictors of mortality

Multivariable logistic regression (Table 2) was performed in 1,416 patients with complete data (72 deaths, 5.1%). The model achieved a C-statistic of 0.746 (95% CI: 0.670–0.812) with adequate calibration (Hosmer-Lemeshow $\chi^2 = 14.34$, df = 8, p = 0.073; McFadden pseudo-$R^2 = 0.16$). Independent predictors of mortality included: severe dengue classification (OR 2.21, 95% CI: 1.18–4.12, p = 0.013), ICU admission (OR 2.29, 95% CI: 1.23–4.25, p = 0.009), elevated neutrophil-to-lymphocyte ratio (OR 1.09 per unit, 95% CI: 1.04–1.14, p < 0.001), and hypoalbuminemia (OR 0.28 per g/dL, 95% CI: 0.17–0.48, p < 0.001). Age was not independently associated with mortality after adjustment (OR 1.01, 95%

**Table 2. Univariate [A] analysis – Risk factors for mortality and multivariable [B] logistic regression – Predictors of mortality.**

| [A] Risk Factor | Events/Total | OR (95% CI) | p-value |
|---|---|---|---|
| Age ≥ 65 years | 23/224 | 2.66 (1.59-4.44) | <0.001 |
| Male sex | 44/831 | 1.12 (0.7-1.8) | 0.721 |
| Diabetes mellitus | 18/288 | 1.33 (0.77-2.3) | 0.296 |
| Hypertension | 16/277 | 1.19 (0.68-2.11) | 0.543 |
| Cardiac disease | 12/112 | 2.49 (1.3-4.78) | 0.011 |
| Chronic kidney disease | 3/17 | 4.15 (1.17-14.77) | 0.051 |
| Severe/Shock dengue | 51/507 | 4.44 (2.7-7.31) | <0.001 |
| Shock at presentation | 16/18 | 190.78 (42.87-848.96) | <0.001 |
| ICU admission | 33/194 | 6.09 (3.75-9.89) | <0.001 |
| Haemorrhagic manifestations | 19/202 | 2.27 (1.32-3.91) | 0.005 |
| Secondary HLH | 5/57 | 1.86 (0.72-4.81) | 0.206 |
| Hypoalbuminemia (<3.5 g/dL) | 24/101 | 8.14 (4.76-13.92) | <0.001 |
| Severe TCP (<20 × 10³/μL) | 13/141 | 2.1 (1.12-3.92) | 0.025 |
| NLR > 4 | 36/350 | 3.22 (2.01-5.15) | <0.001 |
| Non-CLD liver involvement | 17/589 | 0.43 (0.25-0.74) | 0.002 |
| [B] Variable | Adjusted OR | 95% CI | p-value |
| Age (per year) | 1.01 | 1.00-1.03 | 0.058 |
| Male sex | 1.20 | 0.72-2.01 | 0.489 |
| Diabetes mellitus | 0.72 | 0.37-1.41 | 0.334 |
| Cardiac disease | 0.97 | 0.44-2.15 | 0.942 |
| Severe/Shock dengue | 2.31 | 1.25-4.27 | 0.008 |
| ICU admission | 2.45 | 1.28-4.70 | 0.007 |
| Hypoalbuminemia (<3.5 g/dL) | 4.36 | 2.33-8.14 | <0.001 |
| NLR (per unit) | 1.08 | 1.03-1.14 | 0.002 |
| Non-CLD liver involvement | 0.34 | 0.19-0.62 | <0.001 |

OR = Odds Ratio; CI = Confidence Interval. p-values by Fisher's exact test. Model fit: Pseudo $R^2 = 0.18$, AIC = 506.9, n = 1481. Shock excluded due to quasi-complete separation.

CI: 0.99–1.02, p = 0.33). Shock at presentation demonstrated the strongest univariate association with mortality (88.9% vs 4.0%, OR 190.8, 95% CI: 42.9–849.0); however, quasi-complete separation (only 2/18 shock patients survived) precluded stable multivariable adjustment. This finding was presented as univariate only, with the understanding that shock represents a critical endpoint warranting immediate intensive intervention regardless of other risk factors. Please note that 65 patients had missing albumin values. These patients were excluded under complete case analysis when albumin was added to the model. Albumin was included because it emerged as the strongest biomarker predictor of mortality (OR 0.28 per g/dL, p < 0.001; AUC 0.666), reflecting both nutritional status and capillary leak severity.

### (vi) ICU admission predictors

A multivariable logistic regression model was developed for predicting ICU admission at presentation (Table 3), achieving excellent discrimination with a C-statistic of 0.904 (95% CI: 0.88–0.92). Independent predictors of ICU admission included severe dengue (aOR 86.53, 95% CI: 37.23–201.13, p < 0.001), hypoalbuminemia <3.5 g/dL (aOR 3.69, 95% CI: 2.10–6.47, p < 0.001), cardiac disease (aOR 3.05, 95% CI: 1.68–5.54, p < 0.001), and male sex (aOR 1.66, 95% CI: 1.11–2.50, p = 0.014). Notably, haemorrhagic manifestations were associated with reduced ICU admission (aOR 0.50, 95% CI: 0.33–0.77, p = 0.002), likely reflecting that bleeding complications in dengue often occur in less severe classical dengue rather than in patients with severe dengue requiring intensive care. Diabetes mellitus (aOR 1.29, p = 0.310) and NLR (aOR 0.99 per unit, p = 0.620) were not significant independent predictors after adjustment for other variables.

### (vii) Age-stratified analysis

Age-stratified analysis revealed distinct clinical phenotypes across age groups. Patients were categorized as paediatric (<18 years, n = 159, 10.7%), young adult (18−40 years, n = 465, 31.3%), middle-aged (40−60 years, n = 521, 35.1%), and elderly (≥60 years, n = 339, 22.8%). Elderly patients had the highest mortality (9.1%), contributing 41.3% of all deaths despite representing 22.8% of patients. Shock at presentation was markedly higher in elderly (3.5% vs 0.2–0.8% in other groups). Nonetheless, after multivariable adjustment, age was not an independent mortality predictor; disease severity markers drove outcomes across all ages.

### (viii) Geriatric profile outcomes

Among 1,484 hospitalized dengue patients, 339 (22.8%) were elderly (≥60 years; median age 67, IQR 63–73), demonstrating substantially higher comorbidity burden (69.3% vs 19.5%, p < 0.001) including diabetes (42.5% vs 12.6%),

**Table 3. Multivariable logistic regression – Predictors of ICU admission.**

| Variable | Adjusted OR | 95% CI | p-value |
|---|---|---|---|
| Age (per year) | 1.00 | 0.99–1.02 | 0.458 |
| Male sex | 1.66 | 1.11–2.50 | 0.014 |
| Diabetes mellitus | 1.29 | 0.79–2.10 | 0.310 |
| Cardiac disease | 3.05 | 1.68–5.54 | <0.001 |
| Severe dengue | 86.53 | 37.23–201.13 | <0.001 |
| Hypoalbuminemia (<3.5 g/dL) | 3.69 | 2.10–6.47 | <0.001 |
| NLR (per unit) | 0.99 | 0.94–1.04 | 0.620 |
| Hemorrhagic manifestations | 0.50 | 0.33–0.77 | 0.002 |

*OR, odds ratio; CI, confidence interval; NLR, neutrophil-to-lymphocyte ratio. Model C-statistic: 0.904 (95% CI: 0.88–0.92).*

hypertension (48.1% vs 10.0%), and cardiac disease (18.9% vs 4.2%). Elderly patients experienced more severe disease with higher rates of severe dengue (42.2% vs 31.8%, p<0.001), haemorrhagic manifestations (19.2% vs 12.0%, p=0.001), lower platelet nadirs (median 60 vs 80×10³/µL, p<0.001), and reduced albumin (median 3.9 vs 4.1 g/dL, p<0.001). Clinical outcomes were significantly worse in elderly patients: mortality 9.1% vs 3.8% (p<0.001), ICU admission 16.8% vs 9.3% (p<0.001), mechanical ventilation 2.9% vs 0.6% (p=0.001), and secondary bacterial infection 4.1% vs 0.9% (p<0.001) (Fig 4). Multivariable analysis within the elderly subgroup (n=325, C-statistic 0.764) identified severe dengue (aOR 2.95, 95% CI: 1.21–7.19, p=0.017) and hypoalbuminemia (aOR 0.10 per g/dL, 95% CI: 0.04–0.30, p<0.001) as independent mortality predictors, while comorbidities including cardiac disease and diabetes did not independently predict death after adjustment. Mortality varied by age subgroup: 7.0% (60–64 years), 9.7% (65–69 years), 11.1% (70–74 years), 8.5% (75–79 years), and 15.0% (≥80 years), without significant trend (p=0.321).

### (ix) Cluster-based patient phenotypes

K-means clustering was performed on 1,416 patients using six clinical variables (age, platelet nadir, AST, ALT, albumin, and NLR). Silhouette scores were evaluated for k=2 through k=6: k=2 demonstrated strong structure (silhouette=0.860) while k=4 showed reasonable structure (silhouette=0.255). We selected k=4 to balance statistical validity with clinical interpretability, acknowledging that k=2 provides cleaner separation but less granular phenotyping. Four distinct phenotypes were identified: (1) Mild-Classical (n=573, 40.5%) – younger patients (mean age 28 years) with preserved laboratory parameters and 3.0% mortality; (2) Thrombocytopenic-Leak (n=736, 52.0%) – older patients (mean 55 years) with lowest platelet nadir (59×10³/µL) but only 5.2% mortality; (3) Inflammatory-Systemic (n=101, 7.1%) – characterized by highest NLR≥8.0 (range 8.0–29.3, median 14.3) with 10.9% mortality; and (4) Fulminant (n=6, 0.4%) – multi-organ failure with 100% mortality. Bootstrap stability analysis (n=100 iterations) confirmed cluster robustness (mean adjusted Rand index 0.882, SD 0.092). The Fulminant phenotype, while exhibiting uniform mortality, comprised only 6 patients (0.4%) and should be interpreted as identifying outliers with catastrophic presentations rather than a stable clinical subgroup.

### (x) Gender-based differences and impact of metabolic syndrome

Males (n=831, 56.0%) and females (n=653, 44.0%) showed comparable mortality rates (5.3% vs 4.7%, p=0.72). However, males had significantly higher rates of severe thrombocytopenia (12.0% vs 6.3%, p=0.0002) and ICU admission (13.0% vs 8.6%, OR 1.59, p=0.009). Metabolic syndrome (≥2 of: DM, HTN, dyslipidaemia) was present in 226 patients (15.2%) and was associated with higher secondary infection rates (3.5% vs 1.3%, OR 2.85, p=0.028) but not independently associated with mortality after adjustment.

### (xi) Platelet dynamics and thrombocytopenia

Platelet nadir demonstrated significant association with clinical outcomes. The median platelet drop from admission to nadir was 20×10³/µL (IQR 0–55). Severe thrombocytopenia (<20×10³/µL) occurred in 141 patients (9.5%) and was associated with significantly higher mortality (9.2% vs 4.6%, OR 2.10, p=0.025) and ICU admission (43.3% vs 7.7%, OR 9.18, p<0.001). ROC analysis for platelet nadir predicting mortality yielded AUC 0.588 (Fig 5).

### (xii) Biomarker ROC analysis and machine-learning validation for mortality prediction

ROC analysis evaluated biomarkers for mortality prediction. Albumin demonstrated the best discrimination (AUC 0.666, 95% CI: 0.587–0.739) with optimal cutoff ≤3.8 g/dL (sensitivity 59.7%, specificity 70.1%) (Table 4 and Fig 6). NLR (AUC 0.637, 95% CI: 0.569–0.709; cutoff ≥4.8) and CRP (AUC 0.630, 95% CI: 0.562–0.698; cutoff ≥20.4 mg/L) showed acceptable discrimination. Platelet-based markers performed modestly: nadir platelet AUC 0.588 (cutoff ≤36×10³/µL) and admission platelet AUC 0.553 (cutoff ≤61×10³/µL). Ferritin demonstrated limited utility (AUC 0.539).

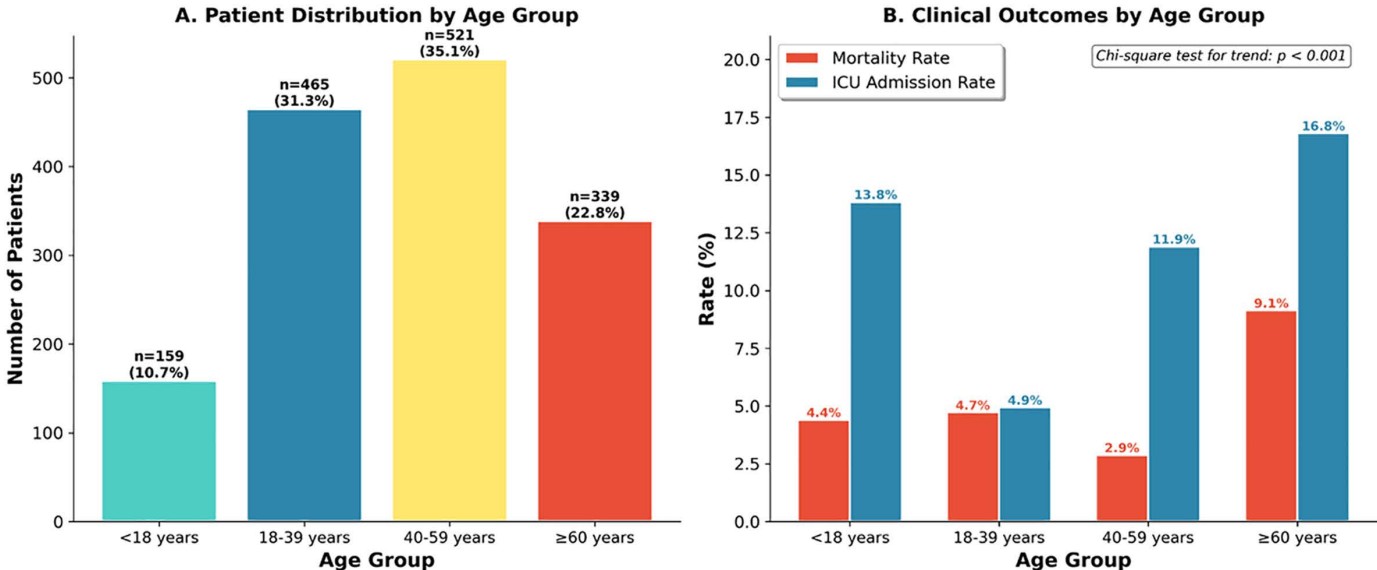

**Fig 4. Age-stratified analysis. Panel A:** Distribution: Paediatric (<18y) 159 (10.7%), Young adult (18-39y) 465 (31.3%), Middle-aged (40-59y) 521 (35.1%), Elderly (≥60y) 339 (22.8%). **Panel B:** Mortality/ICU rates: Paediatric 4.4%/13.8%, Young adult 4.7%/4.9%, Middle-aged 2.9%/11.9%, Elderly 9.1%/16.8%. Chi-square p < 0.001.

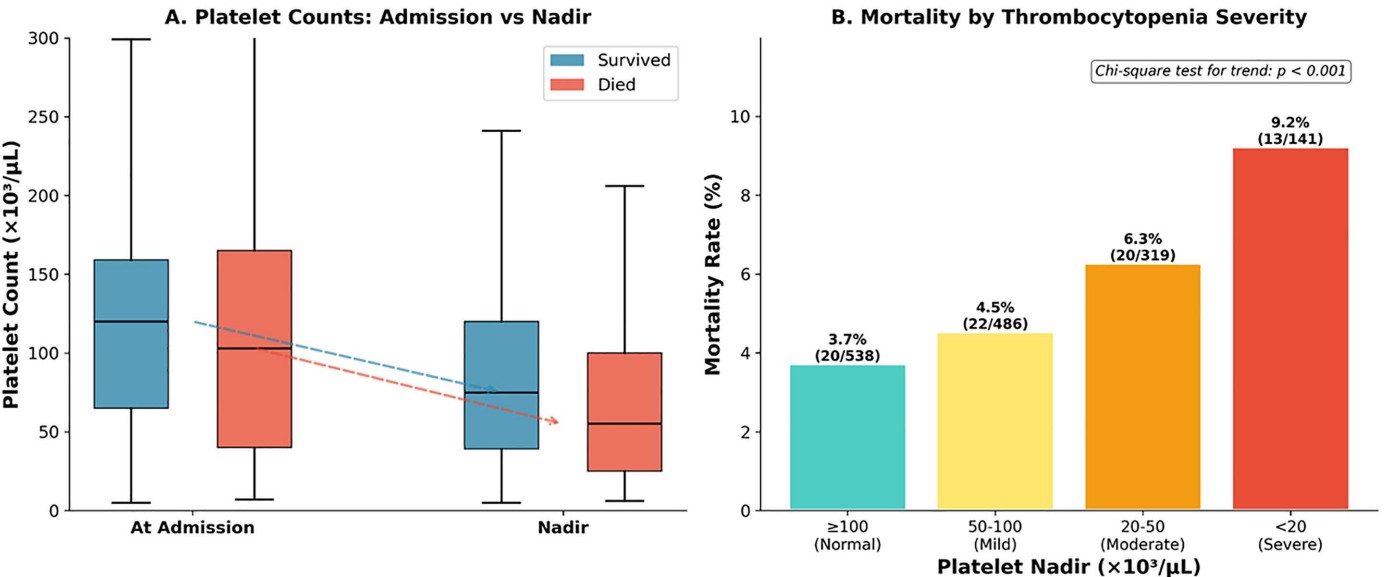

**Fig 5. Platelet dynamics analysis. Panel A:** Platelet trajectories. Survived: admission median 120, nadir 75 × 10³/μL. Died: admission 103, nadir 55 × 10³/μL. **Panel B:** Mortality by thrombocytopenia severity: Normal (≥100) 20/538 (3.7%), Mild (50-100) 22/486 (4.5%), Moderate (20-50) 20/319 (6.3%), Severe (<20) 13/141 (9.2%). Chi-square p < 0.001.

Three machine learning algorithms were compared using 5-fold stratified cross-validation (n = 1,416; 72 deaths). Logistic regression achieved the highest mean AUC (0.718, SD 0.101), followed by Random Forest (0.661, SD 0.114) and Gradient Boosting (0.648, SD 0.072). Variable importance analysis (Random Forest, 1,000 bootstrap iterations) identified

**Table 4. ROC analysis – Biomarkers for mortality prediction.**

| Biomarker | AUC | Optimal Cutoff | Sensitivity (%) | Specificity (%) |
|---|---|---|---|---|
| Albumin (g/dL) | 0.666 | 3.8 | 59.7 | 70.1 |
| NLR | 0.637 | 4.8 | 45.3 | 82.4 |
| CRP (mg/L) | 0.630 | 20.4 | 50.7 | 73.5 |
| Platelet nadir (×10³/μL) | 0.588 | 36 | 41.3 | 75.8 |
| Ferritin (ng/mL) | 0.539 | 2012.1 | 44.3 | 67.7 |
| INR | 0.803 | 1.2 | 70.2 | 81.9 |
| Creatinine (mg/dL) | 0.647 | 1.2 | 41.9 | 79.0 |
| WBC (×10³/μL) | 0.685 | 4 | 73.3 | 56.4 |

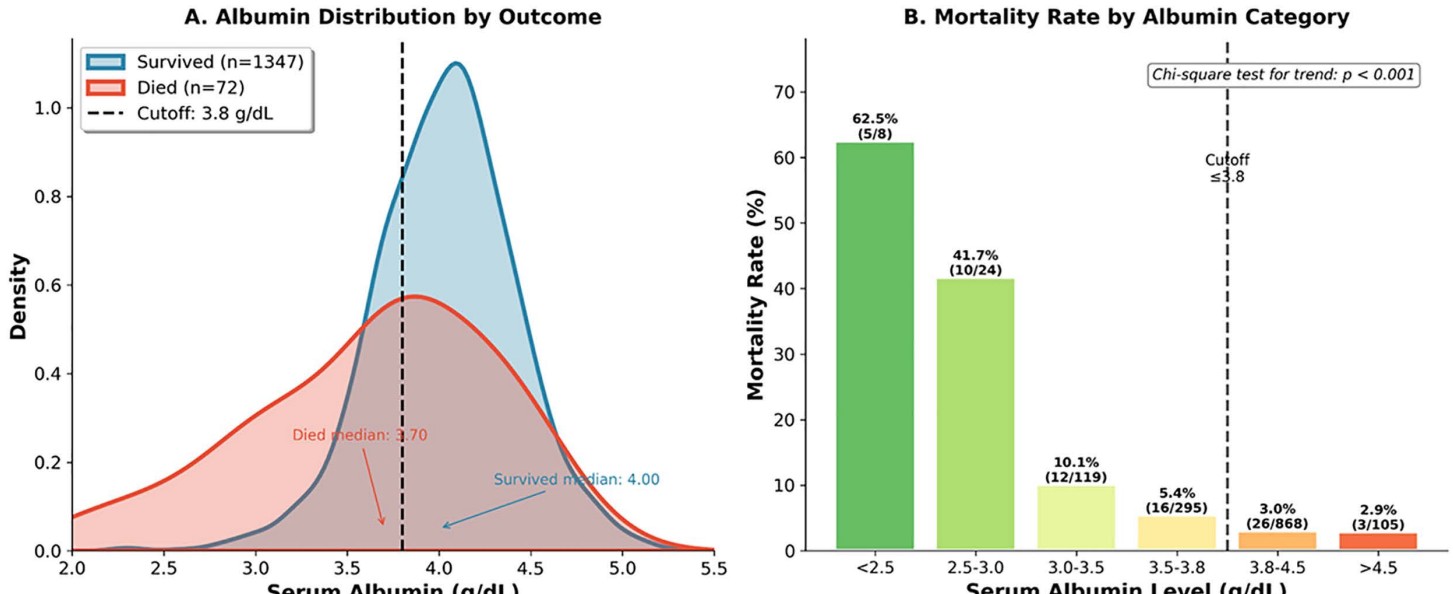

**Fig 6. Albumin as a prognostic marker. Panel A:** Kernel density plots. Survived (n = 1,347): median 4.00 g/dL. Died (n = 72): median 3.70 g/dL. Cutoff 3.8 g/dL marked. **Panel B:** Mortality by category: < 2.5 g/dL 5/8 (62.5%), 2.5-3.0 10/24 (41.7%), 3.0-3.5 12/119 (10.1%), 3.5-3.8 16/295 (5.4%), 3.8-4.5 26/868 (3.0%), > 4.5 3/105 (2.9%). Chi-square p < 0.001.

AST and NLR as top predictors (importance 0.161, 95% CI 0.134–0.195 and 0.157, 95% CI 0.128–0.187 respectively), followed by albumin (0.152), ALT (0.144), and age (0.130) (Fig 7).

To translate the identified independent predictors into a clinically applicable tool, a simple weighted Dengue Severity Risk Score (DeSRS) was developed. The score assigns points based on rounded adjusted odds ratios: severe/shock dengue classification (2 points), hypoalbuminemia <3.5 g/dL (4 points), NLR ≥ 4.8 (2 points), and De Ritis ratio ≥2.0 (2 points), yielding a total range of 0–10. The DeSRS demonstrated good discriminative ability (AUC 0.754) and clinically meaningful risk stratification across four categories: Low risk (score 0–1, n = 672, mortality 1.8%), Moderate risk (score 2–3, n = 540, mortality 3.9%), High risk (score 4–5, n = 187, mortality 8.6%), and Very High risk (score ≥6, n = 85, mortality 30.6%). Among patients scoring ≥8 (n = 36), mortality reached 50.0%. The score demonstrates a 17-fold gradient in mortality from lowest to highest risk categories, using four parameters routinely available at the bedside.

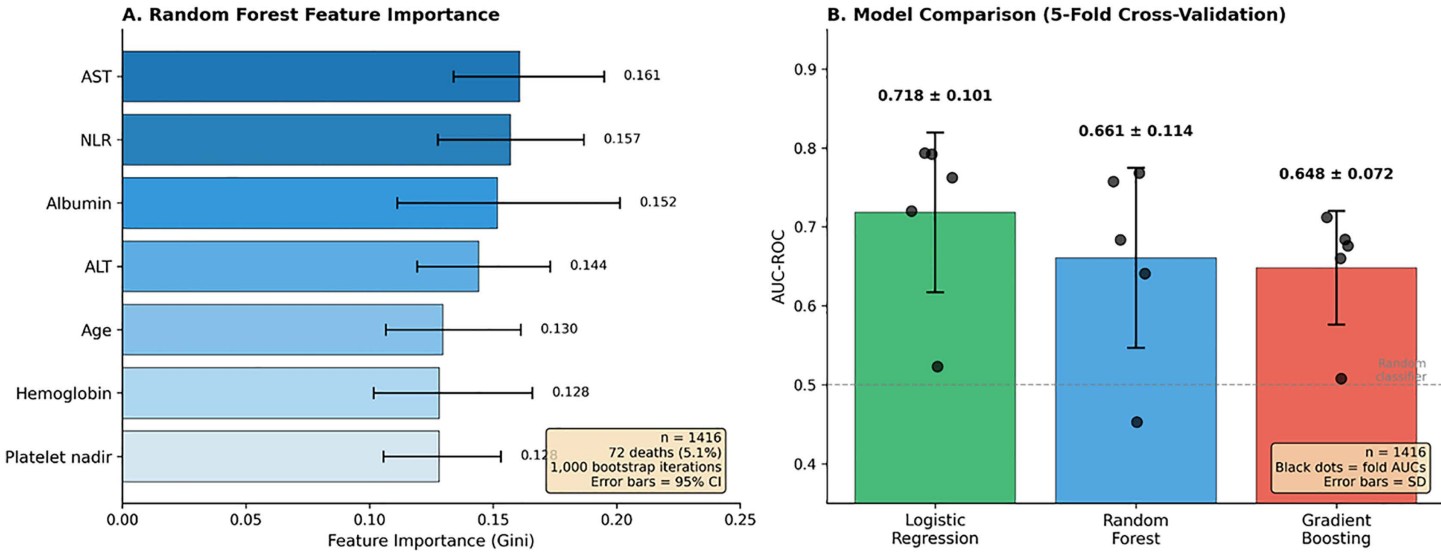

**Fig 7. Machine learning analysis for mortality prediction in dengue fever (n = 1,416 with complete data; 72 deaths, 5.1%). (A)** Random Forest feature importance (Gini importance) with 95% confidence intervals from 1,000 bootstrap iterations. AST (0.161, 95% CI 0.134–0.195), NLR (0.157, 95% CI 0.128–0.187), and albumin (0.152, 95% CI 0.111–0.201) were identified as top predictors. **(B)** Model comparison using 5-fold stratified cross-validation. Logistic regression achieved the highest discriminative performance (AUC 0.718±0.101), followed by Random Forest (0.661±0.114) and Gradient Boosting (0.648±0.072). Black dots represent individual fold AUC values; error bars represent standard deviation.

## (xiii) Organ-specific involvement and comorbidities

Hepatic involvement. Among 1,450 patients without pre-existing CLD, hepatic involvement showed a spectrum of severity: no liver involvement (n = 479, 33.0%), abnormal LFT (n = 563, 38.8%), acute hepatitis (n = 389, 26.8%), acute hepatitis with jaundice (n = 5, 0.3%), and multi-organ failure (MOF; n = 14, 1.0%). Clinical parameters worsened progressively: median age 38–60 years, severe dengue 24.6% to 100%, AST 37–1,883 U/L, platelet nadir 101–22 x10$^3$/μL (all p < 0.001). A significant mortality pattern emerged: no liver involvement 4.8%, abnormal LFTs 3.7%, acute hepatitis 3.6%, but MOF 100% (14/14). MOF was characterized by severe transaminitis (median AST 1,883 U/L), shock (71.4%), concurrent renal failure (creatinine 2.0 mg/dL), and median survival of 1 day. AST > 1000 U/L identified a critical threshold with 53.3% mortality (OR 24.5 vs AST <=1000) and 93.3% ICU admission, warranting immediate intensive care (Fig 8).

Among 1,484 hospitalized dengue patients, three groups were identified based on pre-existing hepatic status: no pre-existing liver disease (n = 861, 58.0%), non-CLD steatotic liver disease (n = 589, 39.7%), and chronic liver disease (CLD; n = 34, 2.3%). Steatotic liver patients were older (median 50 vs 35 years, p < 0.001) with higher diabetes prevalence (27.9% vs 12.8%, p < 0.001) and demonstrated higher transaminase levels (AST 88 vs 64.5 U/L, p < 0.001) and lower platelet nadirs (61 vs 84 x10^3/uL, p < 0.001). Despite this, steatotic liver was paradoxically associated with lower mortality (2.8% vs 6.4%, p = 0.005) and remained independently protective in multivariable analysis (aOR 0.40, 95% CI 0.21–0.74, p = 0.004). To address potential confounding, three complementary causal inference approaches were employed. Propensity score matching (229 pairs; covariates: age, sex, diabetes, hypertension, cardiac disease, severe dengue) confirmed significantly lower mortality in the steatotic group (1.7% vs 10.0%, adjusted OR 0.147, 95% CI 0.049–0.443, p < 0.001). Inverse probability of treatment weighting yielded consistent results (OR 0.326, 95% CI 0.222–0.478, p < 0.001). An expanded multivariable model additionally adjusting for albumin and NLR confirmed the independent protective association (aOR 0.335, 95% CI 0.184–0.609, p < 0.001). CLD patients (73.5% MASLD, 26.5% ALD) had the highest mortality

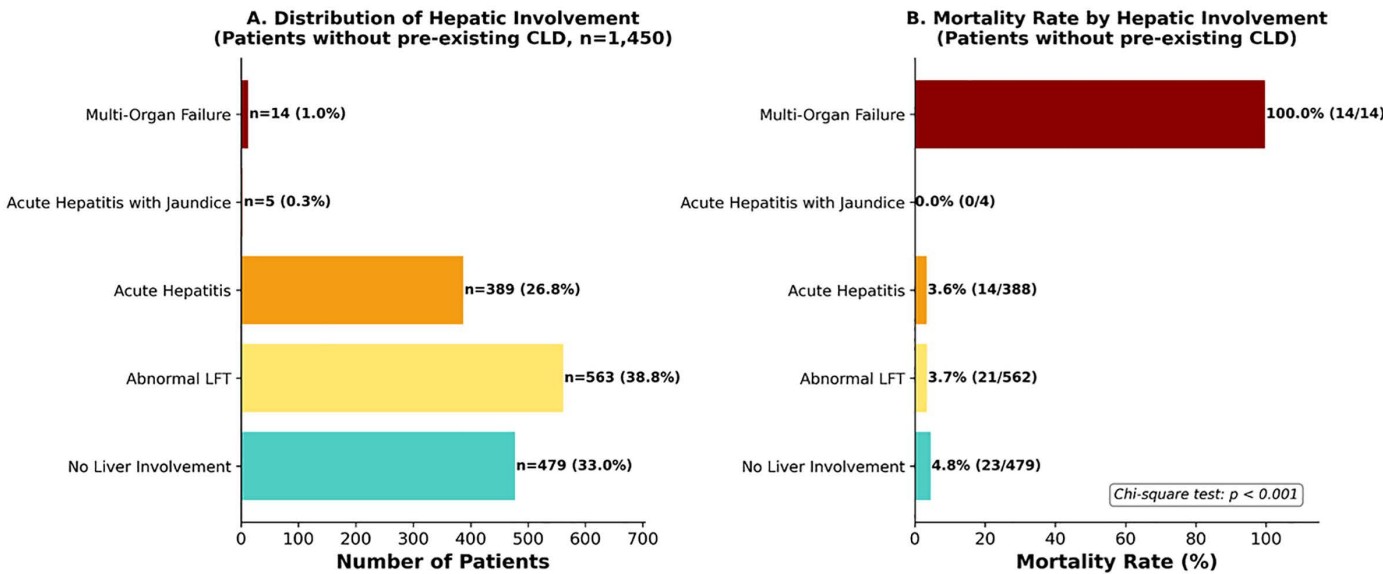

**Fig 8. Hepatic involvement analysis. Note:** Analysis restricted to patients without pre-existing chronic liver disease (n = 1,450) to avoid confounding. **Panel A:** No liver involvement 479 (33.0%), Abnormal LFT 563 (38.8%), Acute Hepatitis 389 (26.8%), Acute Hepatitis with Jaundice 5 (0.3%), Multi-Organ Failure 14 (1.0%). **Panel B:** Mortality: No involvement 23/479 (4.8%), Abnormal LFT 21/563 (3.7%), Acute Hepatitis 14/389 (3.6%), Acute Hepatitis with Jaundice 0/5 (0.0%), **MOF 14/14 (100.0%)**. Chi-square p < 0.001.

(8.8%), with striking differences by compensation status: compensated cirrhosis 0% mortality versus decompensated 23.1%. Haemorrhagic manifestations were more common in CLD (26.5% vs 12.4%, p = 0.038).

Renal involvement. AKI, defined as serum creatinine >1.5 mg/dL, developed in 42 patients (2.9%) among those without pre-existing CKD. AKI was strongly associated with mortality (26.2% vs 4.3%, OR 7.93, 95% CI: 3.81–16.53, p < 0.001). After multivariable adjustment for age, disease severity, and inflammatory markers, AKI remained independently associated with mortality (aOR 3.51, 95% CI: 1.52–8.09, p = 0.003). AKI patients also had significantly higher ICU admission rates (40.5% vs 10.0%, p < 0.001). Dialysis requirement (n = 8) was universally fatal (100% mortality). Pre-existing CKD was present in 17 patients (1.1%). CKD patients had higher mortality (17.6% vs 4.9%) though this did not reach statistical significance (p = 0.051). ICU admission was significantly higher in CKD patients (29.4% vs 10.8%, OR 3.43, p = 0.032). Most notably, CKD was strongly associated with secondary bacterial infection (29.4% vs 1.3%, OR 31.75, p < 0.001)

### (xiv) Secondary hemophagocytic lymphohistiocytosis in Dengue fever

Secondary HLH was identified in 57 patients (3.8%) based on hyperferritinemia with compatible clinical features. HLH patients had dramatically higher ICU admission (45.6% vs 9.7%, OR 7.83, p < 0.001) and secondary infection (10.5% vs 1.3%, OR 9.21, p < 0.001). Ferritin demonstrated excellent discrimination for HLH (AUC 0.837) with optimal threshold 3,535 ng/mL (sensitivity 73.2%, specificity 83.8%). Notably, HLH was not an independent mortality predictor after adjustment for underlying severe dengue (Fig 9).

### (xv) Determinants of prolonged hospitalization

Prolonged hospitalization (>7 days) occurred in 113 of 1,409 survivors (8.0%). Multivariable logistic regression identified independent predictors of prolonged stay (model C-statistic 0.708). Secondary bacterial infection was the strongest

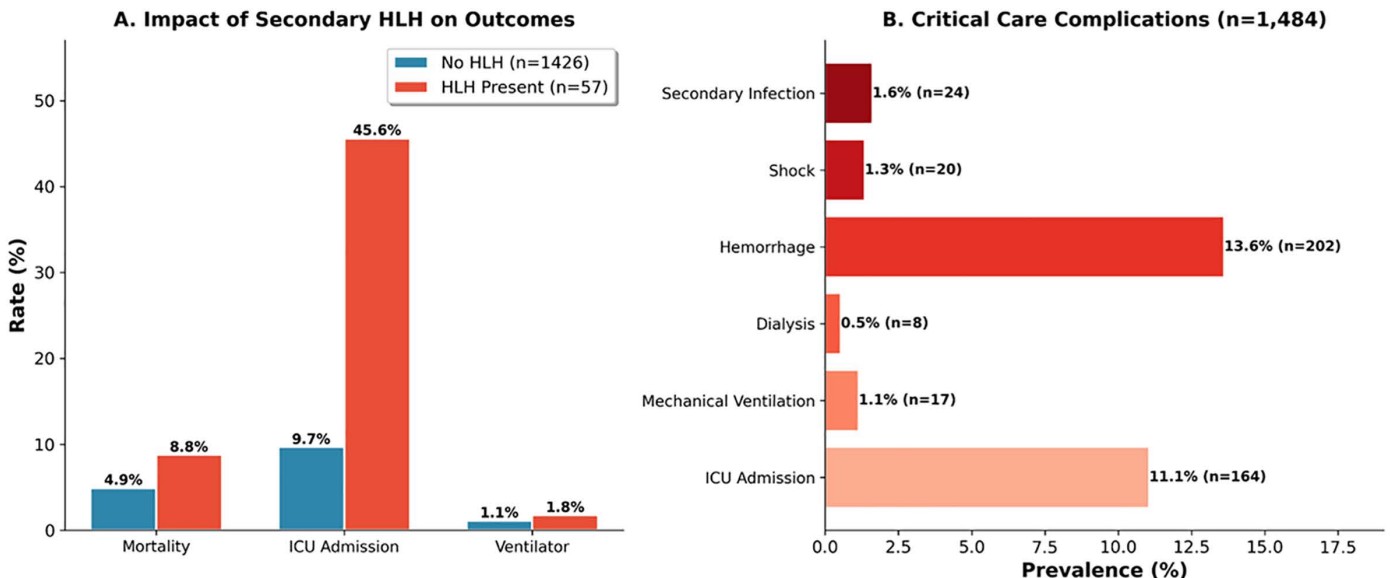

**Fig 9. Secondary HLH and critical care complications. Panel A:** HLH impact (n = 57 with HLH, n = 1,426 without): Mortality 8.8% vs 4.9%, ICU admission 45.6% vs 9.7%, Ventilator 1.8% vs 1.1%. **Panel B:** Complications: ICU 164 (11.1%), Haemorrhage 202 (13.6%), Secondary infection 24 (1.6%), Shock 20 (1.3%), Ventilator 17 (1.1%), Dialysis 8 (0.5%).

predictor (aOR 21.59, 95% CI: 6.79–68.61, p < 0.001), followed by cardiac disease (aOR 2.32, 95% CI: 1.28–4.23, p = 0.006), severe dengue (aOR 1.88, 95% CI: 1.10–3.21, p = 0.022), and age (aOR 1.02 per year, 95% CI: 1.00–1.03, p = 0.017). Among the 24 patients with secondary bacterial infections, bloodstream infections predominated (58.3%), followed by urinary tract (12.5%) and respiratory tract infections (12.5%). Gram-negative organisms were most frequently isolated, including *Escherichia coli* (n = 3), *Klebsiella pneumoniae* (n = 1), *Acinetobacter baumannii* (n = 1), and *Pseudomonas* species (n = 1). Gram-positive organisms included *Staphylococcus aureus* (n = 2). Fungal infections (*Candida* species n = 3, *Aspergillus fumigatus* n = 2) were notable, particularly in ICU patients. Patients with secondary infections had substantially higher mortality (29.2% vs 4.7%) and prolonged hospitalization (median 9 vs 4 days). ICU admission showed a trend toward prolonged stay but did not reach statistical significance (aOR 1.80, 95% CI: 0.97–3.32, p = 0.061) (Fig 10).

**(xvi) Discharge feasibility and readmission analysis**

Of 1,484 patients, 1,409 (94.9%) were discharged alive with an in-hospital mortality rate of 5.1% (75/1,484). Among survivors, median hospital length of stay was 4 days (IQR 3–6), with 28.1% discharged within 3 days, 63.9% between 4–7 days, and 8.0% requiring prolonged hospitalization beyond 7 days. During follow-up (median 152 days, IQR 13–424), 171 patients (12.1%) required readmission. Readmissions were predominantly non-infection-related (114/171, 66.7%), while infection-related readmissions occurred in 56 patients (32.7%). Among infection-related readmissions, respiratory tract infections were most common (37/56, 66.1%), comprising lower respiratory tract infections (n = 34) and upper respiratory tract infections (n = 3), followed by bloodstream infections (12/56, 21.4%). The majority of readmitted patients (145/171, 84.8%) were subsequently discharged, with readmission mortality of 1.8% (3/171). Diabetes (25.1% vs 18.3%, OR 1.50, p = 0.038) and pre-existing cardiac disease (13.5% vs 6.2%, OR 2.34, p = 0.001) were significantly associated with increased readmission risk. Longer index hospitalization also predicted readmission (median 5 vs 4 days, p < 0.001). At last follow-up, cumulative mortality was 5.8% (86/1,484), representing 11 additional post-discharge deaths (0.8% of survivors). The majority achieved stable outcomes (76.1%) or full recovery (16.0%) (Fig 11).

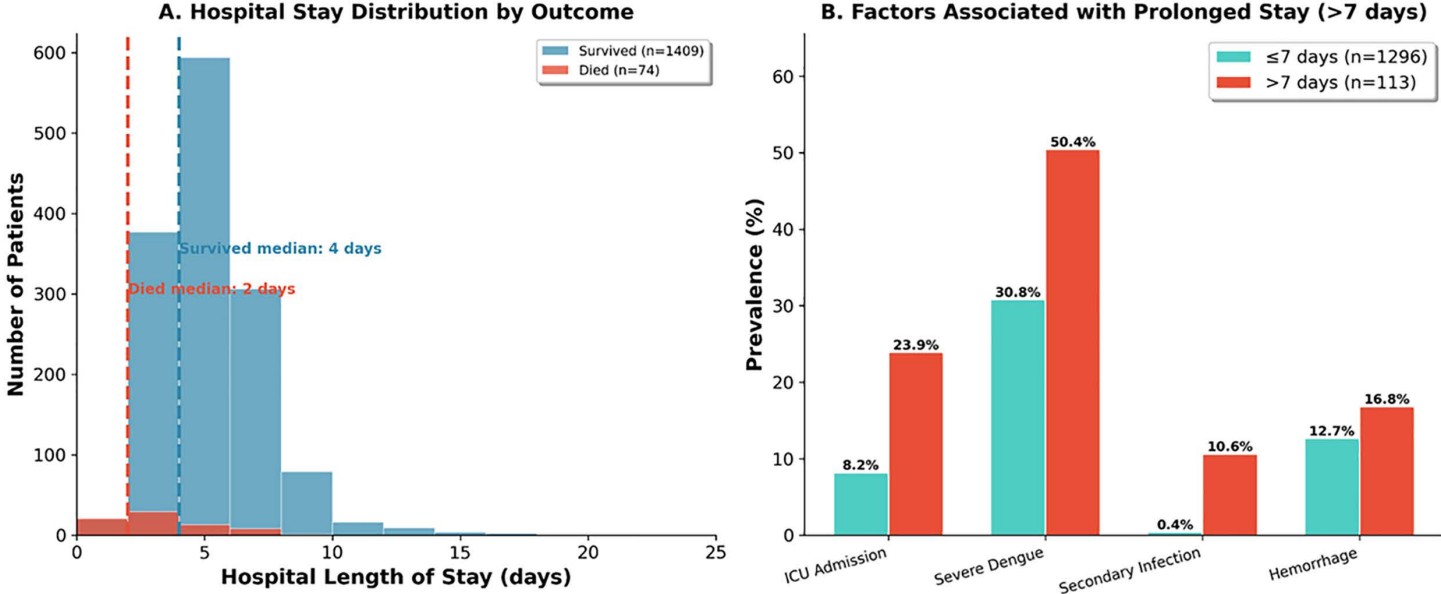

**Fig 10. Hospital stay analysis. Panel A:** Length of stay. Survived (n = 1,409): median 4 days (IQR 3-6). Died (n = 75): median 2 days (IQR 1-5). **Panel B:** Factors in prolonged stay (>7 days, n = 113 vs ≤7 days, n = 1,296): ICU 23.9% vs 8.2%, Severe dengue 50.4% vs 30.8%, Secondary infection 10.6% vs 0.4%, Haemorrhage 16.8% vs 12.7%.

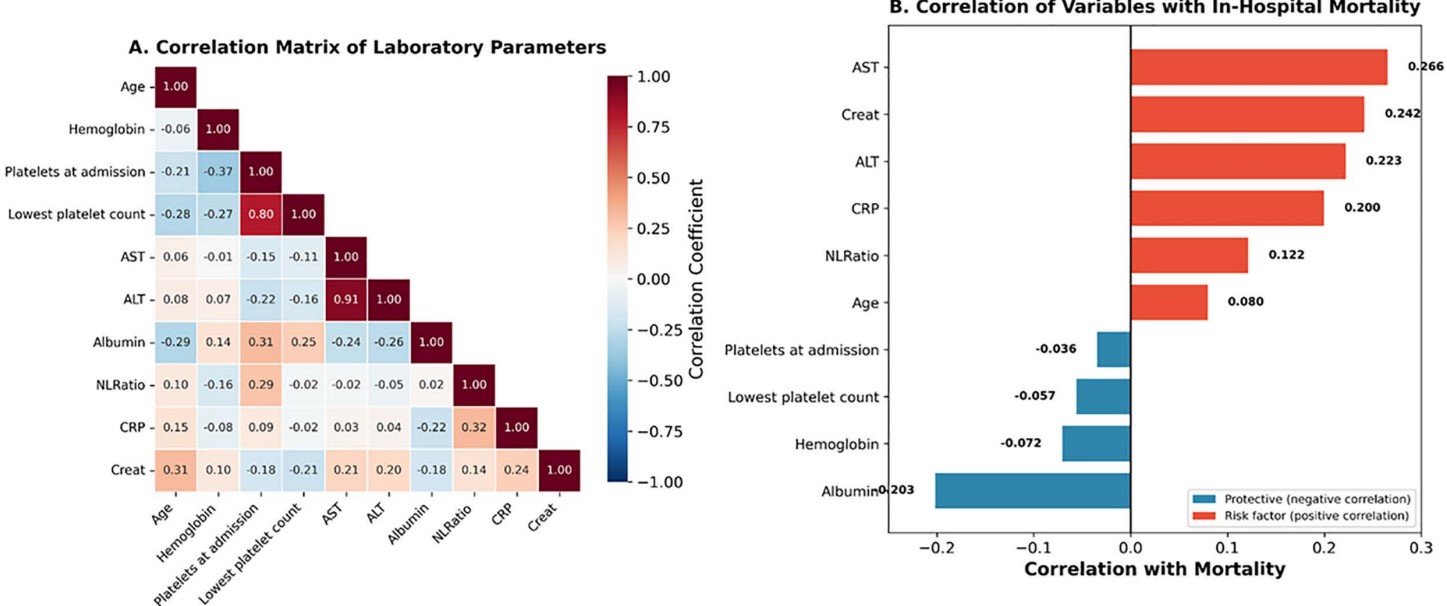

**Fig 11. Correlation analysis. Panel A:** Lower triangular correlation matrix. Notable: AST-ALT r = 0.91, Platelets admission-nadir r = 0.80. **Panel B:** Spearman correlations with mortality (sorted): AST r = 0.266, Creatinine r = 0.242, ALT r = 0.223, CRP r = 0.200, NLR r = 0.122, Age r = 0.080, Platelets admission r = −0.036, Platelets nadir r = −0.057, Hemoglobin r = −0.072, Albumin r = −0.203.

Additionally, among 1,145 patients with long term follow-up data (median 149 days, IQR 13–418), 37 deaths occurred during the observation period. Kaplan-Meier analysis demonstrated overall survival rates of 99.5% at 7 days, 99.0% at 30 days, 97.8% at 90 days, and 95.5% at 365 days. Survival differed significantly by disease severity: classical dengue showed 97.2% one-year survival versus 91.7% for severe/shock dengue (log-rank $\chi^2 = 12.39$, p = 0.0004). Cox proportional hazards regression confirmed severe dengue as an independent predictor of long-term mortality (HR 2.72, 95% CI: 1.41–5.22, p = 0.003), while albumin showed protective effects (HR 0.29 per g/dL, 95% CI: 0.16–0.56, p < 0.001). Age was not significantly associated with survival (HR 1.00, 95% CI: 0.98–1.02, p = 0.92). Schoenfeld residual testing identified violation of the proportional hazards assumption for albumin (p = 0.024); a sensitivity analysis using albumin-stratified Cox regression confirmed that severe dengue remained significantly associated with mortality (HR 2.92, 95% CI: 1.49–5.74, p = 0.002).

### (xvii) Comorbidity-severity interaction analysis

We evaluated effect modification between diabetes mellitus and dengue severity on mortality. Among non-diabetic patients, severe dengue increased mortality 3.76-fold (9.2% vs 2.6%, OR 3.76, 95% CI: 2.16–6.53), while among diabetic patients this increase was 8.64-fold (13.2% vs 1.7%, OR 8.64, 95% CI: 2.44–30.57). Despite a ratio of ORs of 2.30 suggesting potential synergy, the formal interaction term was not statistically significant (OR 2.38, 95% CI: 0.60–9.48, p = 0.218), likely reflecting limited power given few deaths in diabetic subgroups. A similar pattern was observed for ICU admission, with severe dengue increasing ICU risk from 0.6% to 28.0% in non-diabetics and from 0.6% to 42.1% in diabetics. For age and hypoalbuminemia, among 1,419 patients with albumin data, 101 (7.1%) had hypoalbuminemia (<3.5 g/dL). Hypoalbuminemia was significantly associated with mortality across all age groups with similar effect sizes (OR 5.8–12.2), indicating additive rather than synergistic effects. The formal interaction term confirmed no significant effect modification (OR 0.96, p = 0.950). Notably, elderly patients (>60 years) with severe hypoalbuminemia (<3.0 g/dL) had extremely high mortality (80.0%, 4/5), identifying a critical high-risk subgroup requiring aggressive supportive care.

### (xviii) Temporal analysis of Dengue fever - Monthly, seasonal, and year-specific patterns

Among 1,484 hospitalized dengue patients (February 2021–August 2024), case distribution demonstrated marked seasonality with monsoon months (June–September) accounting for 61.6% of admissions and peak volumes in June–July (45.5%, n = 675). Despite this seasonal clustering, disease severity and mortality did not differ significantly across seasons (severe dengue $\chi^2 = 2.73$, p = 0.436; mortality $\chi^2 = 1.95$, p = 0.583), indicating that clinical vigilance should remain constant year-round. Concerning temporal trends emerged over the study period: severe dengue increased from 19.0% (2021) to 47.3% (2024) (Spearman $\rho = 0.213$, p < 0.001), while mortality quadrupled from 1.6% to 6.8% ($\rho = 0.067$, p = 0.010). Concurrent increases in haemorrhagic manifestations (6.3% to 19.3%), plasma leakage indicators, and worsening thrombocytopenia (median platelet nadir 90–66 × $10^3$/μL) accompanied these trends. The year 2024 demonstrated significantly higher mortality compared to 2021–2023 (6.8% vs 4.2%, OR 1.66, 95% CI: 1.04–2.65, p = 0.034), with calendar year showing a trend toward independent mortality prediction after adjustment for disease severity and albumin (aOR 1.43, 95% CI: 0.99–2.09, p = 0.059). Year-specific multivariable models confirmed that severe dengue classification (2023: aOR 2.67, p = 0.008; 2024: aOR 2.69, p = 0.022) and hypoalbuminemia (2023: aOR 0.31, p = 0.001; 2024: aOR 0.21, p < 0.001) remained consistent mortality predictors across years, supporting their utility for risk stratification regardless of temporal context. Notably, August demonstrated the highest monthly mortality (9.6%) despite being a post-peak month, suggesting delayed complications warrant extended monitoring beyond the acute transmission period (Fig 12).

### (xix) Clinical scoring systems

We validated multiple prognostic scoring systems in 1,484 hospitalized dengue patients. The Albumin-Bilirubin (ALBI) score demonstrated excellent correlation with hepatic involvement severity, with median scores progressively worsening

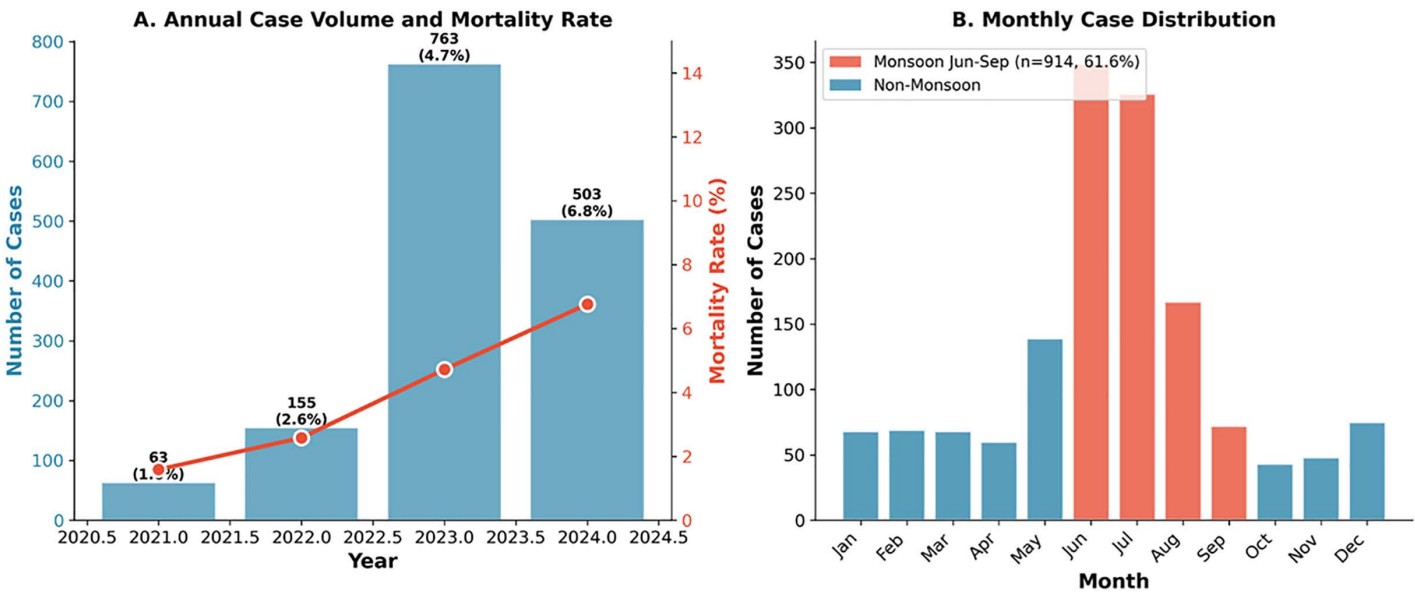

**Fig 12. Temporal trends. Panel A:** Annual cases: 2021 (n = 63, 1.6% mortality), 2022 (n = 155, 2.6%), 2023 (n = 763, 4.7%), 2024 (n = 503, 6.8%). **Panel B:** Monthly distribution. Monsoon months (June-September): 914/1,484 (61.6%).

from −2.87 in patients without liver involvement to −1.51 in multi-organ failure. ALBI grade showed a striking dose-response relationship with outcomes: ICU admission rates increased from 6.3% (Grade 1) to 21.2% (Grade 2) and 50.0% (Grade 3), while mortality rose from 3.1% to 8.1% to 70.0% across grades (AUC 0.670 for mortality, 0.726 for ICU prediction). Among 164 ICU patients (19.0% mortality), SAPS-3 substantially outperformed SOFA for mortality prediction (AUC 0.852 vs 0.652). Non-survivors had significantly higher SAPS-3 scores than survivors (median 70 vs 49, p < 0.001), with an optimal threshold of ≥62 providing 71.0% sensitivity and 93.9% specificity. In multivariable analysis combining SOFA, SAPS-3, and ALBI (C-statistic 0.874), SAPS-3 remained strongly predictive (aOR 1.16 per point, 95% CI: 1.09–1.23, p < 0.001) while SOFA lost significance (aOR 0.86, p = 0.325), indicating SAPS-3 should be preferred for dengue ICU prognostication. The De Ritis ratio (AST/ALT), reflecting the characteristic AST-predominant hepatitis pattern of dengue (median 1.26, IQR 0.97–1.67), demonstrated a dose-response relationship with mortality: 5.6% at ratio ≥1.0, increasing to 13.1% at ≥2.5 (AUC 0.639). A De Ritis ratio ≥2 independently predicted mortality after adjustment for age, severe dengue, and albumin (aOR 2.25, 95% CI: 1.26–4.00, p = 0.006). These findings support the use of ALBI for hepatic risk stratification, SAPS-3 over SOFA for ICU mortality prediction, and De Ritis ratio ≥2 as an additional marker warranting enhanced monitoring (Fig 13).

## Discussion

This large retrospective cohort study of 1,484 hospitalized dengue patients provided comprehensive insights into mortality predictors, hepatic involvement patterns, and prognostic scoring system performance in a contemporary South Asian population. The overall in-hospital mortality rate of 5.1% and ICU admission rate of 11.1% align with reported ranges from endemic regions, though direct comparisons were complicated by heterogeneous case definitions and admission thresholds across studies. Our analysis identified several clinically actionable findings: hypoalbuminemia emerged as the strongest biomarker predictor of mortality, the ALBI grade demonstrated excellent prognostic stratification, SAPS-3 substantially outperformed SOFA for ICU mortality prediction, and steatotic liver disease was paradoxically associated with improved survival – a finding with important implications for clinical risk assessment.

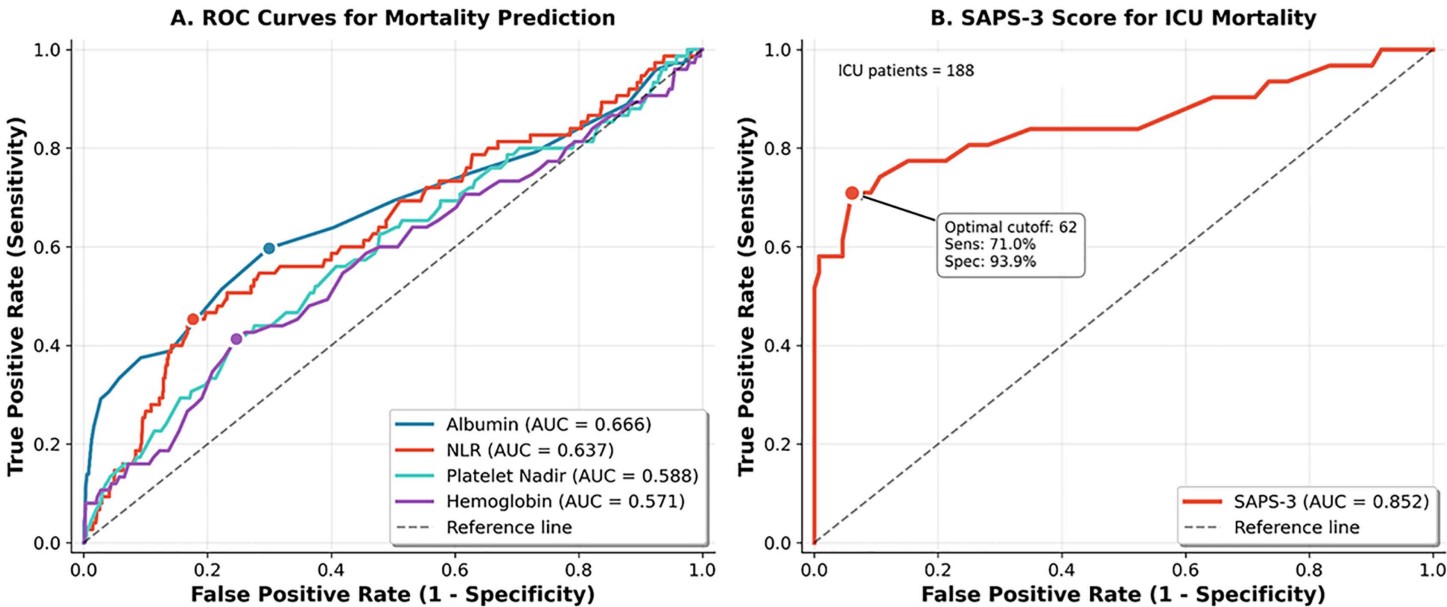

**Fig 13. ROC curves for mortality prediction. Panel A:** Biomarker AUCs: Albumin 0.666, NLR 0.637, Platelet nadir 0.588, Hemoglobin 0.571. **Panel B:** SAPS-3 in ICU patients (n = 188): AUC 0.852, optimal cutoff 62, sensitivity 71.0%, specificity 93.9%.

The independent predictors of mortality identified in our multivariable model – severe dengue classification, ICU admission, elevated neutrophil-to-lymphocyte ratio (NLR), and hypoalbuminemia – were biologically plausible and largely consistent with existing literature. A recent systematic review and meta-analysis identified severe hepatitis, dengue shock syndrome, altered mental status, and diabetes mellitus as key mortality risk factors [16]. Our findings extend this evidence by demonstrating that albumin, a readily available and inexpensive biomarker, provides the best single-variable discrimination for mortality prediction in this cohort. Hypoalbuminemia in dengue reflects both the severity of capillary leak syndrome and underlying nutritional/inflammatory status, making it a composite marker of disease severity [17]. The optimal cutoff of ≤3.8 g/dL offers a practical threshold for risk stratification at the bedside. Notably, shock at presentation demonstrated an extraordinarily high univariate odds ratio with near-complete mortality, confirming that dengue shock syndrome remains the most critical prognostic determinant warranting immediate intensive intervention regardless of other risk factors [6].

The NLR has gained recognition as an accessible inflammatory biomarker across diverse clinical contexts, from sepsis to malignancy [18]. In our dengue cohort, NLR independently predicted mortality with an optimal cutoff of ≥4.8. This finding complements emerging evidence on inflammatory and endothelial markers in dengue prognostication. Vuong et al. demonstrated that combinations of inflammatory (IL-8, ferritin, IL-1RA) and vascular markers (syndecan-1, angiopoietin-2) measured during the febrile phase predict progression to severe dengue [19]. Similarly, Sivasubramanian et al. identified IL-6, TNFR1, thrombomodulin, and angiopoietin-2 as promising prognostic markers for dengue severity [20]. While these specialized biomarkers may offer superior discrimination, NLR has the practical advantage of derivation from routine complete blood count, making it immediately applicable in resource-limited settings where dengue burden is highest.

Perhaps the most intriguing finding of our study is the apparent protective effect of pre-existing steatotic liver disease on dengue mortality. This observation contradicts traditional teaching that obesity and metabolic dysfunction predispose to severe dengue through chronic pro-inflammatory states [7–9]. However, this finding aligns with the "obesity paradox" increasingly recognized in critical care literature, wherein moderate adiposity may confer survival advantage in acute illness through enhanced metabolic reserves, adipokine-mediated immunomodulation, and potential endotoxin

neutralization by circulating lipoproteins [11,13]. Karampela et al. comprehensively reviewed this phenomenon in critical illness and sepsis, noting that meta-analyses consistently demonstrate obesity-related survival benefits despite methodological limitations [12]. The paradoxically lower mortality observed in patients with steatotic liver disease could be explained by differential susceptibility to two distinct death phenotypes identified in this cohort. Deaths followed either a shock/catastrophic phenotype (characterized by multi-organ failure, hemodynamic collapse, ventilator/dialysis requirement) or a plasma leak phenotype (characterized by third-space fluid accumulation without organ failure, predominantly non-ICU deaths). Steatotic liver patients were markedly protected from the shock phenotype: only 2/577 (0.3%) developed this presentation compared to 27/873 (3.1%) in patients without pre-existing liver disease, with 0% versus 81.5% mortality respectively. Critically, within the plasma leak phenotype, mortality was similar between groups (4.0% vs 2.8%, p = 0.274), indicating that the entire mortality difference is attributable to the 22 shock phenotype deaths occurring exclusively in patients without pre-existing liver disease. This protection against catastrophic deterioration persisted even among patients with severe transaminitis (AST > 1000 U/L): steatotic patients showed 0% progression to MOF (0/6) versus 88.9% (8/9) in those without pre-existing disease, despite equivalent biochemical severity. These findings suggest that hepatic steatosis confers specific protection against the inflammatory cascade leading to shock and multi-organ failure – possibly through preserved hepatic synthetic reserve (reflected by higher albumin: 3.7 vs 3.0 g/dL in severe cases) and immune modulation from chronic low-grade hepatic inflammation attenuating cytokine storm. Our study represents, to our knowledge, the first demonstration of this paradox specifically in tropical viral infection. Whether this reflects true biological protection or unmeasured confounding (such as earlier presentation or lower admission thresholds in metabolically compromised patients) cannot be definitively resolved with retrospective data. Nonetheless, this finding has practical implications: the presence of steatotic liver disease should not automatically escalate risk assessment, and clinicians should avoid assuming worse outcomes in this population.

In contrast to steatotic liver disease, CLD, particularly decompensated cirrhosis, was associated with markedly adverse outcomes. The mortality differential between compensated (0%) and decompensated cirrhosis (23.1%) underscores the critical importance of hepatic reserve in determining dengue outcomes. Kulkarni et al. previously demonstrated that cirrhotic patients lack classical dengue features (such as haemoconcentration) and have prolonged hospitalization and higher mortality [5]. Our findings confirm and extend these observations in a larger cohort with explicit stratification by liver disease severity status. The population-based Taiwan cohort study similarly identified liver cirrhosis as a risk factor for hospitalization, ICU admission, and mortality in dengue patients [21]. The mechanistic basis likely involves cirrhosis-associated immune dysfunction, impaired synthetic function affecting coagulation factor production, and reduced hepatic clearance of inflammatory mediators – all potentially exacerbating the systemic inflammatory response characteristic of severe dengue.

The spectrum of hepatic injury we observed – ranging from abnormal liver tests (38.8%) through acute hepatitis (26.8%) to multi-organ failure (1.0%), reflects the heterogeneous pathophysiology of dengue hepatitis. The striking mortality gradient, from 3.6–4.8% in milder presentations to 100% in multi-organ failure, aligns with prior reports [22]. Critically, we identified AST > 1000 U/L as a threshold warranting immediate ICU consideration, with 53.3% mortality and 93.3% ICU admission in this subgroup. This finding corroborates data from Teerasarntipan et al., who validated MELD and ALBI scores for predicting acute liver failure and mortality in dengue-induced severe hepatitis [23]. The AST-predominant transaminitis pattern characteristic of dengue (median De Ritis ratio 1.26 in our cohort) differs from the ALT-predominant pattern typical of viral hepatitis, reflecting the distinct pathophysiology involving both hepatocyte necrosis and myocyte injury [24]. Our finding that De Ritis ratio ≥2 independently predicted mortality (adjusted OR 2.25) adds prognostic utility to this routinely calculated parameter.

A key contribution of our study was the comparative validation of prognostic scoring systems in a large dengue cohort. The ALBI grade demonstrated excellent dose-response relationships with both ICU admission (6.3% to 50.0% across grades 1–3) and mortality (3.1% to 70.0%), supporting its utility for hepatic risk stratification in acute infectious contexts

beyond primary liver cancer. Recent studies have validated ALBI in diverse settings including cirrhosis with sepsis and acute kidney injury, demonstrating that this objective measure of hepatic synthetic function has broad prognostic applicability [25]. In our ICU subgroup, SAPS-3 substantially outperformed SOFA for mortality prediction (AUC 0.852 vs 0.652). This differential performance is notable given that SOFA was specifically designed for sepsis-related organ dysfunction assessment, while SAPS-3 incorporates admission characteristics and comorbidities. Zhu et al. similarly demonstrated SAPS-3 superiority over SOFA for 28-day mortality prediction in sepsis patients [26]. Our findings suggest that in dengue, where thrombocytopenia is near-universal and thus contributes minimally to discrimination, SAPS-3 should be preferred for ICU prognostication. The optimal SAPS-3 threshold of ≥62 (sensitivity 71.0%, specificity 93.9%) provides a practical decision-support tool for intensive care resource allocation.

The unsupervised clustering analysis identified four distinct patient phenotypes with differential mortality risk, ranging from 3.0% in the Mild-Classical cluster to 100% in the Fulminant cluster. This phenotypic heterogeneity has important implications for clinical trial design and personalized management. The Inflammatory-Systemic cluster, characterized by highest NLR despite moderate thrombocytopenia, exhibited 10.9% mortality and may represent a subgroup warranting immunomodulatory intervention. The Inflammatory-Systemic phenotype was primarily defined by NLR ≥ 8.0 providing a practical bedside threshold for identification. Patients meeting this criterion could be considered for enhanced monitoring regardless of age. Our machine learning comparison demonstrated that traditional logistic regression performed comparably to ensemble methods (Random Forest, Gradient Boosting) for mortality prediction, suggesting that complex algorithms offer limited advantage over interpretable linear models in this context. This finding has practical implications for clinical deployment, as logistic regression coefficients can be directly translated into bedside risk scores. The internal-external validation using temporal splitting (2021–2023 training, 2024 testing) confirmed model stability across the study period despite increasing mortality rates. Accordingly, machine learning analysis demonstrated acceptable discriminative ability for mortality prediction (AUC 0.718), indicating the model (Logistic Regression) correctly identified higher-risk patients approximately 72% of the time. Variable importance analysis revealed that readily available, inexpensive laboratory parameters – specifically hepatic transaminases (AST, ALT), NLR, and serum albumin – alongside patient age were the strongest predictors of mortality. These findings suggest that routine monitoring of liver function, inflammatory markers, and nutritional status can help clinicians identify high-risk dengue patients who may benefit from intensified monitoring or early ICU referral.

Building on the multivariable analysis, we developed a simple weighted Dengue Severity Risk Score (DeSRS) incorporating four bedside-available parameters: dengue severity classification, albumin, NLR, and De Ritis ratio. The score demonstrated good discrimination (AUC 0.754) with a 17-fold mortality gradient across risk categories, outperforming any individual biomarker. Importantly, all score components are derived from routine clinical assessment and standard laboratory panels, requiring no specialized assays or computational tools. While our DeSRS performed comparably to published dengue prognostic scores that typically require more complex inputs, it must be emphasized that this score was derived from a single-center retrospective cohort and requires prospective, multi-center external validation before clinical deployment. Validation across diverse endemic regions with differing serotype circulation, healthcare infrastructure, and patient demographics will be essential to establish generalizability.

The temporal analysis revealed concerning trends: severe dengue increased from 19.0% (2021) to 47.3% (2024), mortality quadrupled from 1.6% to 6.8%, and haemorrhagic manifestations rose from 6.3% to 19.3%. These trends may reflect evolving viral characteristics, changing serotype dominance, increasing secondary infections in the population, or shifts in healthcare-seeking behaviour and admission thresholds. Calendar year showed a near-significant independent association with mortality after adjustment (adjusted OR 1.43, p = 0.059), suggesting that factors beyond disease severity classification are contributing to worsening outcomes. Whether these trends represent local epidemiological dynamics or broader regional patterns warrants investigation through genomic surveillance and multi-center collaboration.

Several secondary findings merit clinical attention. Secondary hemophagocytic lymphohistiocytosis (HLH) was identified in 3.8% of patients and was associated with dramatically increased ICU admission (45.6% vs 9.7%) and secondary infection rates. Dengue-associated HLH represents a recognized but under-appreciated complication, with recent reviews emphasizing the need for heightened clinical suspicion in patients with hyperferritinemia and clinical deterioration [27,28]. Our ferritin threshold of 3,535 ng/mL for HLH identification (sensitivity 73.2%, specificity 83.8%) provides practical guidance, though ferritin measurement was not universally available, limiting generalizability. Acute kidney injury, present in 2.9% of patients without pre-existing CKD, was strongly associated with mortality (26.2% vs 4.3%), and dialysis requirement was universally fatal. These findings underscore the importance of renal monitoring and early intervention in dengue patients with declining urine output or rising creatinine in the context of renal involvement that can precede clinical deterioration [29].

Elderly patients (≥60 years) constituted 22.8% of our cohort but contributed 41.3% of all deaths, with mortality of 9.1% compared to 3.8% in younger patients. This disproportionate mortality burden was driven by higher comorbidity prevalence, more severe disease presentation, and greater haemorrhagic manifestations. However, in age-stratified multivariable analysis, severe dengue classification and hypoalbuminemia – not age per se – emerged as independent mortality predictors, suggesting that disease severity rather than chronological age drives outcomes. This aligns with the recent published literature which demonstrated that dengue disease severity drove higher mortality outcomes in the elderly compared to younger patients [30]. This finding has important implications for triage: elderly patients without severe disease features should not be reflexively escalated to intensive monitoring based on age alone, while younger patients with severe features warrant equivalent attention.

Our study has several limitations that warrant acknowledgment. First, the retrospective single-center design introduces potential selection bias and limits generalizability to other endemic regions with different healthcare infrastructure, dengue serotype circulation, and population characteristics. Second, missing data for certain variables may have influenced results, though sensitivity analyses suggested robustness to missing data patterns. Third, we lacked viral serotype data and could not assess the contribution of secondary versus primary infection to outcomes – a critical determinant of dengue severity. Fourth, the definition of steatotic liver disease relied on imaging reports without histological confirmation, potentially introducing misclassification. Fifth, the "obesity paradox" finding, while statistically robust, may reflect unmeasured confounding rather than true biological protection, and should be interpreted cautiously. Sixth, the *Fulminant* phenotype cluster comprised only six patients, limiting its reliability as a stable clinical subgroup. Finally, long-term follow-up was available for only a subset of patients, and post-discharge outcomes may be incomplete.

Nonetheless, several directions for future research emerge from our findings. Prospective validation of the identified prognostic factors – particularly the albumin cutoff of ≤3.8 g/dL, SAPS-3 threshold of ≥62, and ALBI grade stratification – in independent cohorts would strengthen their clinical applicability. The paradoxical protective effect of steatotic liver disease warrants mechanistic investigation through adipokine profiling and metabolomic analysis to determine whether this represents true biological protection or confounding. Multi-center studies incorporating viral serotype data and genetic polymorphisms would enhance understanding of host-pathogen interactions driving severe outcomes. Development and validation of a composite prognostic score integrating albumin, NLR, disease severity classification, and De Ritis ratio could provide a practical bedside tool superior to any individual parameter. Finally, the temporal trends toward increasing severity warrant ongoing epidemiological surveillance and investigation of potential contributory factors.

## Conclusion

In conclusion, in our study, the current largest single center evaluation of hospitalized dengue infection, comprehensive analysis of 1,484 patients identified hypoalbuminemia as the strongest biomarker predictor of mortality, validated the ALBI grade for hepatic risk stratification in acute dengue, demonstrated SAPS-3 superiority over SOFA for ICU prognostication, and revealed a paradoxical protective association of steatotic liver disease with survival. Decompensated cirrhosis, acute

kidney injury, and secondary HLH confer substantially increased mortality risk. Four distinct patient phenotypes with differential outcomes were identified through unsupervised clustering. These findings provide practical guidance for risk stratification and resource allocation in dengue-endemic settings, while highlighting areas requiring further investigation.

## Supporting information

**S1 Text. Missing variables dataset.** Word document representing all the missing data for all key analytical variables of patients included in the study.
(DOCX)

## Author contributions

**Conceptualization:** Aryalakshmi Sreemohan, Cyriac Abby Philips.

**Data curation:** Aryalakshmi Sreemohan, Arif Hussain Theruvath, Ambily Baby.

**Formal analysis:** Arif Hussain Theruvath, Cyriac Abby Philips.

**Methodology:** Aryalakshmi Sreemohan, Arif Hussain Theruvath, Ambily Baby, Cyriac Abby Philips, Tharun Tom Oommen.

**Resources:** Ambily Baby.

**Supervision:** Tharun Tom Oommen, Santhichandra Pai, Salini Baby John, Jaicob Varghese, Rizwan Ahamed, Ajit Tharakan, Philip Augustine.

**Validation:** Tharun Tom Oommen, Santhichandra Pai, Salini Baby John, Jaicob Varghese, Rizwan Ahamed, Ajit Tharakan, Philip Augustine.

**Writing – original draft:** Cyriac Abby Philips.

**Writing – review & editing:** Aryalakshmi Sreemohan, Arif Hussain Theruvath, Ambily Baby, Cyriac Abby Philips, Tharun Tom Oommen, Santhichandra Pai, Salini Baby John, Jaicob Varghese, Rizwan Ahamed, Ajit Tharakan, Philip Augustine.

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
