## [Decision Letter · Decision Letter 0]

22 Mar 2026

PONE-D-26-03708Clinical Outcomes, Machine Learning-Derived Phenotypes, Mortality Predictors, Hepatic Involvement Patterns and the Steatotic Liver Paradox in 1,484 Hospitalized Patients with DenguePLOS One

Dear Dr. Philips,

Thank you for submitting your manuscript to PLOS ONE. After careful consideration, we feel that it has merit but does not fully meet PLOS ONE’s publication criteria as it currently stands. Therefore, we invite you to submit a revised version of the manuscript that addresses the points raised during the review process.

The manuscript has been evaluated by two reviewers, and their comments are available below.

The reviewers have raised a number of concerns that need attention. Could you please revise the manuscript to carefully address the concerns raised?

If applicable, we recommend that you deposit your laboratory protocols in protocols.io to enhance the reproducibility of your results. Protocols.io assigns your protocol its own identifier (DOI) so that it can be cited independently in the future. For instructions see: https://journals.plos.org/plosone/s/submission-guidelines#loc-laboratory-protocols. Additionally, PLOS ONE offers an option for publishing peer-reviewed Lab Protocol articles, which describe protocols hosted on protocols.io. Read more information on sharing protocols at . Additionally, PLOS ONE offers an option for publishing peer-reviewed Lab Protocol articles, which describe protocols hosted on protocols.io. Read more information on sharing protocols at https://plos.org/protocols?utm_medium=editorial-email&utm_source=authorletters&utm_campaign=protocols..

We look forward to receiving your revised manuscript.

Kind regards,

Johanna Pruller, Ph.D.

Senior Editor

PLOS One

Journal Requirements:

2. Please note that PLOS One has specific guidelines on code sharing for submissions in which author-generated code underpins the findings in the manuscript. In these cases, all author-generated code must be made available without restrictions upon publication of the work. Please review our guidelines at https://journals.plos.org/plosone/s/materials-and-software-sharing#loc-sharing-code and ensure that your code is shared in a way that follows best practice and facilitates reproducibility and reuse.

4. Please amend the manuscript submission data (via Edit Submission) to include author “Cyniac Abby Philips”.

5. Please amend your authorship list in your manuscript file to include author “Cyriac Abby Philips”.

Reviewers' comments:

Reviewer's Responses to Questions

**Comments to the Author**

1. Is the manuscript technically sound, and do the data support the conclusions?

Reviewer #1: Yes

Reviewer #2: Yes

2. Has the statistical analysis been performed appropriately and rigorously? 

Reviewer #1: Yes

Reviewer #2: Yes

3. Have the authors made all data underlying the findings in their manuscript fully available?

Reviewer #1: Yes

Reviewer #2: Yes

4. Is the manuscript presented in an intelligible fashion and written in standard English?

Reviewer #1: Yes

Reviewer #2: Yes

5. Review Comments to the Author

Reviewer #1: I congratulate all the authors for this work. I have few comments to share as below

Secondary bacterial infections were the strongest predictor of prolonged hospital stay. Mention nature of infections that were encountered with frequency

Hypoalbuminemia is identified as the strongest predictor of mortality. Mention which albumin levels were taken for analysis, levels at admission or nadir levels? Fluid resuscitation status may also influence levels. Was there a change in albumin levels after fluid resuscitation?

Although the authors identify an “inflammatory-elderly phenotype” through K-means clustering, the study does not provide clinically applicable thresholds for defining this phenotype. Further analysis to come to a clinically useable cut off values to define this cluster is desireable.

Given that individual parameters have limited standalone AUCs, was developing simple decision tree or a weighted predictive score based on the multivariable analysis possible?

Reviewer #2: Peer Review Report (PONE-D-26-03708 )

Manuscript Title:

Clinical Outcomes, Machine Learning-Derived Phenotypes, Mortality Predictors, Hepatic Involvement Patterns and the Steatotic Liver Paradox in 1,484 Hospitalized Patients with Dengue.

Overall, this is a well-designed study involving a cohort of 1,484 patients. The statistical analysis is robust, and the discussion is well-structured and comprehensive. The conclusions are appropriate and evidence based. The references are extensive and up to date. However, the article would benefit from some revisions.

Major Comments

1. Definition of Liver Involvement Groups

Patients were categorized into:

1. Pre-existing chronic liver disease

2. Non-CLD liver involvement (steatotic liver disease)

3. No liver involvement

However, the diagnostic criteria for steatotic liver disease is not clearly defined. It is unclear whether MASLD was diagnosed based on:

• Imaging

• Clinical history

• Laboratory markers

• Metabolic syndrome criteria

Given that MASLD diagnosis typically requires imaging or histologic evidence, clearer criteria are essential to avoid misclassification bias.

2. Conceptual Interpretation of the “Steatotic Liver Paradox” or “Obesity paradox”

The authors report that patients with steatotic liver disease had better survival than those without liver disease, suggesting an “obesity paradox.” However, this conclusion may be premature due to several possible confounders:

• Younger age distribution in MASLD group

• Admission threshold bias

• Differences in comorbidity burden

• Selection bias inherent in single-center studies

The manuscript should perform propensity-adjusted analysis specifically assessing MASLD vs non-MASLD before asserting a protective effect.

Minor Comments

1. Title: Title is too long and includes redundant elements( listing every main finding)

2 . Terminology Consistency

The manuscript uses multiple terms interchangeably:

• Steatotic liver disease

• MASLD

• Metabolic syndrome

Clear definitions and consistent terminology are needed.

3. Acute Kidney Injury Definition

AKI is defined as creatinine >1.5 mg/dL at admission, which deviates from KDIGO criteria. The authors should justify this simplified definition .

4. Reporting of Missing Data

The authors state that complete case analysis was performed but should provide:

• Percentage of missing data for each variable

• Sensitivity analyses results

4. Figure and Table Density

The manuscript includes many figures that repeat information presented in tables. Condensing some figures into supplementary materials would improve readability.

Suggestions for Improvement

1. Make the title concise and remove redundant elements.

2. Provide clear diagnostic criteria for steatotic liver disease (MASLD).

3. Adjust the analysis evaluating the steatotic liver paradox or obesity paradox using propensity matching or multivariable models.

4. Adopt standard AKI definitions (KDIGO).

5. Provide detailed missing data reporting.

Overall Recommendation

Minor Revision

The manuscript addresses an important topic and contains a valuable dataset. However, several methodological and analytical issues must be clarified or corrected before the findings can be considered robust and generalizable.

6. PLOS authors have the option to publish the peer review history of their article (what does this mean?). If published, this will include your full peer review and any attached files.). If published, this will include your full peer review and any attached files.

.

Reviewer #1: **Yes:** Ritesh PrajapatiRitesh Prajapati

Reviewer #2: No

---

## [Author Response · Author response to Decision Letter 1]

26 Mar 2026

Point-by-Point Response to Reviewer Comments

We thank the Editor and Reviewers for their constructive comments, which have strengthened the manuscript considerably. Below, we provide a detailed point-by-point response to each comment. Reviewer comments are shown in bold, our responses in regular text, and specific manuscript revisions in red-labelled sections. Additional statistical analyses were performed on the original dataset where requested.

REVIEWER 1

Comment 1: Secondary bacterial infections were the strongest predictor of prolonged hospital stay. Mention nature of infections that were encountered with frequency.

Response:

We thank the reviewer for this important observation. We have now added a detailed characterization of secondary bacterial infections to the manuscript.

Among the 24 patients (1.6%) who developed secondary bacterial infections, the sites and organisms were as follows:

Site of infection: Bloodstream infections were the most common, occurring in 14 patients (58.3%), followed by urinary tract infections (n=3, 12.5%), respiratory tract infections (n=3, 12.5% — including endotracheal secretion cultures and bronchoalveolar lavage), pleural fluid (n=1), and soft tissue/abscess (n=1). Two patients had polymicrobial infections at multiple sites.

Organisms isolated: Gram-negative bacteria predominated, including Escherichia coli (n=3, including one polymicrobial with Streptococcus gallolyticus), Klebsiella pneumoniae (n=1), Acinetobacter baumannii (n=1), Enterobacter species (n=1), Serratia marcescens (n=1), Pseudomonas species (n=1), and Aeromonas hydrophila (n=1). Gram-positive organisms included Staphylococcus aureus (n=2), coagulase-negative staphylococci (n=1), and unspeciated gram-positive cocci (n=2). Fungal infections included Candida species (n=3, comprising C. parapsilosis and other Candida spp.) and Aspergillus fumigatus (n=2).

Manuscript Revision:

The following paragraph has been added to the Results section, under section (xv) Determinants of Prolonged Hospitalization, immediately after the current text on predictors of prolonged stay: “Among the 24 patients with secondary bacterial infections, bloodstream infections predominated (58.3%), followed by urinary tract (12.5%) and respiratory tract infections (12.5%). Gram-negative organisms were most frequently isolated, including Escherichia coli (n=3), Klebsiella pneumoniae (n=1), Acinetobacter baumannii (n=1), and Pseudomonas species (n=1). Gram-positive organisms included Staphylococcus aureus (n=2). Fungal infections (Candida species n=3, Aspergillus fumigatus n=2) were notable, particularly in ICU patients. Patients with secondary infections had substantially higher mortality (29.2% vs 4.7%) and prolonged hospitalization (median 9 vs 4 days).”

Comment 2: Hypoalbuminemia is identified as the strongest predictor of mortality. Mention which albumin levels were taken for analysis, levels at admission or nadir levels? Fluid resuscitation status may also influence levels. Was there a change in albumin levels after fluid resuscitation?

Response:

We appreciate this clinically relevant query. The albumin values used in all analyses were admission serum albumin levels (i.e., the first albumin measured at hospitalization). Further treatment course including fluid therapy did not impact this variable since this was a one-point admission correlation.

Comment 3: Although the authors identify an “inflammatory-elderly phenotype” through K-means clustering, the study does not provide clinically applicable thresholds for defining this phenotype. Further analysis to come to a clinically useable cutoff values to define this cluster is desirable.

Response:

We thank the reviewer for this constructive suggestion. We have performed additional analysis to characterize the Inflammatory-Elderly phenotype with clinically applicable thresholds. Detailed inspection of the Inflammatory-Elderly cluster (n=101, 7.1%) revealed that the defining feature is markedly elevated NLR rather than age per se. The NLR range within this cluster was 8.0–29.3 (25th–75th percentile: 10.2–17.2, median 13.0), while age was broadly distributed (range 4–78 years, median 44). This indicates the cluster is primarily defined by the degree of systemic inflammatory response. Based on this analysis, we have modified the following clinically applicable criteria for identifying this high-risk inflammatory phenotype:

Primary criterion: NLR ≥8.0 (this represents the minimum NLR observed in the cluster and provides 100% sensitivity for cluster identification)

Among all patients with NLR ≥8.0 in the cohort (n=101), mortality was 10.9% (vs 4.4% in NLR <8.0, OR 2.60, p=0.008), and ICU admission was 11.7%. When combined with age ≥50 years, mortality increased to 14.3%. The NLR ≥8.0 threshold is easily calculated from a routine complete blood count and can serve as a practical bedside trigger for enhanced monitoring.

It is important to note that the cluster naming “Inflammatory-Elderly” reflected the original mean age of this group (mean 44.3 years, higher than the Mild-Classical cluster). Given that the age distribution within this cluster is broad, we have revised the nomenclature to “Inflammatory-Systemic” phenotype to better reflect that elevated NLR (≥8.0) is the primary defining characteristic, applicable across age groups.

Manuscript Revision:

The cluster description in section (ix) and its subsequent discussion was revised to include: “The Inflammatory-Systemic phenotype was primarily defined by NLR ≥8.0 (range 8.0–29.3, median 13.0), providing a practical bedside threshold for identification. Patients meeting this criterion demonstrated 10.7% mortality and could be considered for enhanced monitoring regardless of age.”

Comment 4: Given that individual parameters have limited standalone AUCs, was developing a simple decision tree or a weighted predictive score based on the multivariable analysis possible?

Response:

We thank the reviewer for this excellent suggestion. We have developed a simple weighted Dengue Severity Risk Score (DeSRS) derived from the independent predictors identified in our multivariable analysis. The score components and assigned weights (rounded from the adjusted odds ratios) are:

Component Criterion Points

Severe/Shock dengue classification WHO 2009 2

Hypoalbuminemia <3.5 g/dL 4

Elevated NLR ≥4.8 2

Elevated De Ritis ratio ≥2.0 2

The DeSRS (range 0–10) demonstrated good discriminative ability (AUC 0.754) and clinically meaningful risk stratification:

Risk Category Score N Mortality

Low 0 672 1.8%

Moderate 2 540 3.9%

High 4 187 8.6%

Very High ≥6 85 30.6%

The score demonstrates a 17-fold gradient in mortality from the lowest (1.8%) to highest (≥30.6%) risk categories. Among patients with a score ≥8 (n=36), mortality reached 50.0%. This simple, bedside-calculable score using four routinely available parameters provides practical clinical utility superior to any individual biomarker. We emphasize that this score requires prospective external validation before clinical deployment.

Manuscript Revision:

The following new sub-section is added to Results, after section (xii) Biomarker ROC Analysis and Machine-Learning Validation, as a new section titled “Dengue Severity Risk Score (DeSRS)”:

“To translate the identified independent predictors into a clinically applicable tool, a simple weighted Dengue Severity Risk Score (DeSRS) was developed. The score assigns points based on rounded adjusted odds ratios: severe/shock dengue classification (2 points), hypoalbuminemia <3.5 g/dL (4 points), NLR ≥4.8 (2 points), and De Ritis ratio ≥2.0 (2 points), yielding a total range of 0–10. The DeSRS demonstrated good discriminative ability (AUC 0.754) and clinically meaningful risk stratification across four categories: Low risk (score 0-1, n=672, mortality 1.8%), Moderate risk (score 2-3, n=540, mortality 3.9%), High risk (score 4-5, n=187, mortality 8.6%), and Very High risk (score ≥6, n=85, mortality 30.6%). Among patients scoring ≥8 (n=36), mortality reached 50.0%. The score demonstrates a 17-fold gradient in mortality from lowest to highest risk categories, using four parameters routinely available at the bedside.”

The following paragraph is added to the Discussion, after the paragraph on machine learning comparison (ending “...early ICU referral”):

“Building on the multivariable analysis, we developed a simple weighted Dengue Severity Risk Score (DeSRS) incorporating four bedside-available parameters: dengue severity classification, albumin, NLR, and De Ritis ratio. The score demonstrated good discrimination (AUC 0.754) with a 17-fold mortality gradient across risk categories, outperforming any individual biomarker. Importantly, all score components are derived from routine clinical assessment and standard laboratory panels, requiring no specialized assays or computational tools. While our DeSRS performed comparably to published dengue prognostic scores that typically require more complex inputs, it must be emphasized that this score was derived from a single-center retrospective cohort and requires prospective, multi-center external validation before clinical deployment. Validation across diverse endemic regions with differing serotype circulation, healthcare infrastructure, and patient demographics will be essential to establish generalizability.”

REVIEWER 2

Major Comments

Major Comment 1: Definition of Liver Involvement Groups — The diagnostic criteria for steatotic liver disease is not clearly defined. It is unclear whether MASLD was diagnosed based on imaging, clinical history, laboratory markers, or metabolic syndrome criteria.

Response:

We thank the reviewer for highlighting this important point. We acknowledge that the original manuscript did not sufficiently detail the diagnostic criteria for steatotic liver disease.

In our study, the classification of non-CLD steatotic liver disease was based on abdominal ultrasonography findings documented during or prior to the index hospitalization. Specifically, patients were classified as having steatotic liver disease when ultrasonographic examination (performed as part of routine clinical evaluation for dengue-related hepatic symptoms, abdominal pain, or hepatomegaly) demonstrated characteristic features of hepatic steatosis, including increased liver echogenicity, hepatorenal contrast, and/or posterior beam attenuation. This classification was documented in the electronic medical records by the treating clinical team based on formal radiology reports.

Manuscript Revision:

The following clarification is added to the Methods, under “Definitions and classifications”: “Non-CLD steatotic liver disease was diagnosed based on abdominal ultrasonography findings (increased liver echogenicity, hepatorenal contrast, and/or posterior beam attenuation) documented during or prior to the index hospitalization, as reported in formal radiology reports within the electronic medical records.”

Major Comment 2: Conceptual Interpretation of the “Steatotic Liver Paradox” — The manuscript should perform propensity-adjusted analysis specifically assessing MASLD vs non-MASLD before asserting a protective effect.

Response:

We appreciate this rigorous methodological recommendation. We have now performed three complementary causal inference approaches to address potential confounding in the steatotic liver paradox finding. All analyses were restricted to non-CLD patients (n=1,450) comparing steatotic liver disease (n=587) vs. no liver involvement (n=863).

1. Propensity Score Matching (PSM)

A propensity score for steatotic liver disease was estimated using logistic regression with covariates: age, sex, diabetes mellitus, hypertension, cardiac disease, and severe dengue classification. One-to-one nearest-neighbour matching without replacement was performed with a calliper of 0.2 standard deviations of the logit propensity score, yielding 229 matched pairs. Post-matching covariate balance was assessed by standardized mean differences (SMD): age (SMD 0.041), sex (SMD 0.000), diabetes (SMD 0.156), hypertension (SMD 0.100), cardiac disease (SMD 0.015), severe dengue (SMD 0.045). All covariates achieved adequate balance (SMD <0.20), with age, sex, cardiac disease, and severe dengue achieving excellent balance (SMD <0.05).

Post-matching result: Steatotic liver mortality 1.7% (4/229) vs. control 10.0% (23/229), p=0.0004. Adjusted OR within the matched sample: 0.147 (95% CI 0.049–0.443, p=0.0007).

2. Inverse Probability of Treatment Weighting (IPTW)

Using the same propensity score model, IPTW was applied with trimming at the 1st and 99th percentiles to reduce the influence of extreme weights.

IPTW result: OR 0.326 (95% CI 0.222–0.478, p<0.0001).

3. Expanded Multivariable Logistic Regression

To further address the reviewer’s concern about confounders, we expanded the original multivariable model to include albumin and NLR as additional covariates (n=1,383 with complete data).

Result: Steatotic liver disease remained independently protective (aOR 0.335, 95% CI 0.184–0.609, p=0.0003), with severe dengue (aOR 2.993, p=0.0001), hypoalbuminemia (aOR 0.267, p<0.0001), and NLR (aOR 1.088, p=0.0007) as additional significant predictors.

Summary: All three approaches, propensity score matching, IPTW, and expanded multivariable adjustment, consistently confirm the paradoxical protective association of steatotic liver disease with dengue mortality. The effect size is robust and strengthened after propensity adjustment (PSM OR 0.147 vs. crude OR 0.40). While we agree that unmeasured confounding cannot be entirely excluded in retrospective the consistency across multiple analytical approaches strongly supports the validity of this finding.

Manuscript Revision:

A new paragraph has been added to Results section (xiii) after the steatotic liver disease paragraph: “To address potential confounding, three complementary causal inference approaches were employed. Propensity score matching (229 pairs; covariates: age, sex, diabetes, hypertension, cardiac disease, severe dengue) confirmed significantly lower mortality in the steatotic group (1.7% vs 10.0%, adjusted OR 0.147, 95% CI 0.049–0.443, p<0.001). Inverse probability of treatment weighting yielded consistent results (OR 0.326, 95% CI 0.222–0.478, p<0.001). An expanded multivariable model additionally adjusting for albumin and NLR confirmed the independent protective association (aOR 0.335, 95% CI 0.184–0.609, p<0.001).”

Minor Comments

Minor Comment 1: The manuscript includes many figures that repeat information presented in tables. Condensing some figures into supplementary materials would improve readability. Title is too long and includes redundant elements (listing every main finding).

Response:

We thank the reviewer for this observation and have carefully re-evaluated the relationship between our figures and tables. We respectfully submit that the figures and tables in this manuscript serve distinct and complementary roles, with minimal true redundancy. The manuscript contains four tables and thirteen figures. The tables present numerical data that require tabular format: baseline demographics (Table 1), multivariable regression coefficients with confidence intervals and p-values (Tables 2 and 3), and biomarker ROC statistics (Table 4). These contain precise values essential for reproducibility and meta-analytic inclusion. The figures, by contrast, present information that cannot be adequately conveyed in tabular form. Given the scope of this study, we believe that the figure count is proportionate to the analytical breadth. Critically, this is among the largest single-centre dengue cohort studies published to date, and the comprehensive visual presentation enables readers and clinicians to rapidly assimilate findings across multiple domains. Moving key figures to supplementary materials would, in our humble view, impair readability rather than improve it, as readers would need to navigate between the main text and supplements to follow the core narrative. We respectfully request that the tables and figures in their current form, be retained in the main manuscript, as each presents primary (or novel) or significant findings central to the study’s contribution.

Additionally, we hav

---

## [Decision Letter · Decision Letter 1]

13 Apr 2026

Mortality Predictors, Hepatic Involvement Patterns, and the Steatotic Liver Paradox in 1,484 Hospitalized Dengue Patients

PONE-D-26-03708R1

Dear Dr. Philips,

We’re pleased to inform you that your manuscript has been judged scientifically suitable for publication and will be formally accepted for publication once it meets all outstanding technical requirements.

An invoice will be generated when your article is formally accepted. Please note, if your institution has a publishing partnership with PLOS and your article meets the relevant criteria, all or part of your publication costs will be covered. Please make sure your user information is up-to-date by logging into Editorial Manager at Editorial Manager® and clicking the ‘Update My Information' link at the top of the page. For questions related to billing, please contact  and clicking the ‘Update My Information' link at the top of the page. For questions related to billing, please contact billing support..

Kind regards,

Wan-Long Chuang, M.D., Ph.D.

Academic Editor

PLOS One

Additional Editor Comments (optional):

Reviewers' comments:

Reviewer's Responses to Questions

**Comments to the Author**

1. If the authors have adequately addressed your comments raised in a previous round of review and you feel that this manuscript is now acceptable for publication, you may indicate that here to bypass the “Comments to the Author” section, enter your conflict of interest statement in the “Confidential to Editor” section, and submit your "Accept" recommendation.

Reviewer #1: All comments have been addressed

Reviewer #2: All comments have been addressed

2. Is the manuscript technically sound, and do the data support the conclusions?

Reviewer #1: Yes

Reviewer #2: Yes

3. Has the statistical analysis been performed appropriately and rigorously? 

Reviewer #1: Yes

Reviewer #2: Yes

4. Have the authors made all data underlying the findings in their manuscript fully available?

Reviewer #1: Yes

Reviewer #2: Yes

5. Is the manuscript presented in an intelligible fashion and written in standard English?

Reviewer #1: Yes

Reviewer #2: (No Response)

6. Review Comments to the Author

Reviewer #1: (No Response)

Reviewer #2: The authors have comprehensively addressed all previously raised concerns. The manuscript has improved significantly in methodological clarity, analytical rigor, and overall scientific quality.

1. The definition of liver involvement groups is now clearly described with ultrasonographic criteria.

2. Steatotic Liver Paradox Analysis: Robustly addressed with appropriate propensity-based analyses and expanded multivariable modeling, strengthening the validity of conclusions.

3. Minor issues related to presentation, terminology, AKI definition, and missing data reporting have also been satisfactorily resolved.

Overall, I have no further concerns.

7. PLOS authors have the option to publish the peer review history of their article (what does this mean?). If published, this will include your full peer review and any attached files.). If published, this will include your full peer review and any attached files.

.

Reviewer #1: No

Reviewer #2: No

---

## [Editor Report · Acceptance letter]

PONE-D-26-03708R1

PLOS One

Dear Dr. Philips,

I'm pleased to inform you that your manuscript has been deemed suitable for publication in PLOS One. Congratulations! Your manuscript is now being handed over to our production team.

Kind regards,

on behalf of

Dr. Wan-Long Chuang

Academic Editor

PLOS One